# Analysis Methods for Diagnosing Rare Neurodevelopmental Diseases with Episignatures: A Systematic Review of the Literature

**DOI:** 10.3390/biomedicines13123043

**Published:** 2025-12-11

**Authors:** Albert Alegret-García, Alejandro Cáceres, Marta Sevilla-Porras, Luís A. Pérez-Jurado, Juan R. González

**Affiliations:** 1Barcelona Institute for Global Health (ISGlobal), 08003 Barcelona, Spain; albert.alegret@isglobal.org (A.A.-G.); alejandro.caceres@isglobal.org (A.C.); 2Centre for Biomedical Network Research on Epidemiology and Public Health (CIBERESP), 08003 Barcelona, Spain; 3Genetics Unit, Department of Medicine and Life Sciences, Universitat Pompeu Fabra, 08003 Barcelona, Spain; marta.sevilla@upf.edu (M.S.-P.); luis.perez@upf.edu (L.A.P.-J.); 4Centre for Biomedical Network Research on Rare Diseases (CIBERER), 08003 Barcelona, Spain; 5Genetics Service, Hospital del Mar and Institut Hospital del Mar d’Investigacions Mèdiques (IMIM), 08003 Barcelona, Spain

**Keywords:** rare disease, neurodevelopmental disorders, DNA methylation, episignatures, DMPs, DMRs, VUS, machine learning

## Abstract

**Background:** Rare diseases (RDs) and neurodevelopmental disorders (NDDs) remain under-researched due to their low prevalence, leaving significant gaps in diagnostic strategies. Beyond next-generation sequencing, epigenetic profiling and particularly episignatures have emerged as a promising complementary diagnostic tool and for reclassifying variants of uncertain significance (VUS). However, clinical implementation remains limited, hindered by non-standardized methodologies and restricted data sharing that impede the development of sufficiently large datasets for robust episignature development. **Methods:** We conducted a systematic literature review following PRISMA 2020 guidelines to identify all studies reporting episignatures published between 2014 and 2025. The review summarizes methodological approaches used for episignature detection and implementation, as well as reports of epimutations. **Results:** A total of 108 studies met the inclusion criteria. All but three employed Illumina methylation arrays, mostly 450 K and EPIC versions for patient sample analysis. Three main methodological phases were identified: data quality control, episignature detection, and classification model training. Despite methodological variability across these stages, most studies demonstrated high predictive capabilities, often relying on methodologies developed by a small number of leading groups. **Conclusions:** Epigenetic screening has significant potential to improve diagnostic yield in RDs and NDDs. Continued methodological refinement and collaborative standardization efforts will be crucial for its successful integration into clinical practice. Nevertheless, key challenges persist, including the need for secure and ethical data-sharing frameworks, external validation, and methodological standardization.

## 1. Introduction

A rare disease (RD) is by definition a disorder that affects less than 1 in 2000 individuals (according to the European Union [1]), or fewer than 200,000 people in total (according to the USA [2]). Despite their individual rarity, over 7000 distinct RDs have been described [1,3], collectively affecting about 5% of the population [4].

Neurodevelopmental disorders (NDDs), many of which also qualify as RDs, encompass a diverse group of conditions characterized by cognitive, behavioral, motor, and adaptive impairments [5]. These disorders arise due to abnormal genetic and/or environmental influences that affect neurodevelopment [6].

Diagnosing NDDs and other RDs is inherently challenging because of overlapping clinical phenotypes, shared biomarkers or genetic variants of uncertain significance (VUS), and the high cost of advanced diagnostic techniques. As a result, it can take years of testing and clinical consultations before reaching a reliable diagnosis, if one is achieved at all [7,8]. This prolonged uncertainty, often referred to as the “diagnostic odyssey,” imposes significant emotional and psychological burdens on patients and their families. Despite extensive efforts, it is estimated that up to 50% of patients remain undiagnosed [9], highlighting the urgent need for innovative diagnostic strategies [10].

When aberrant phenotypes emerge, such as congenital anomalies or developmental delay, there is a need for an early etiological diagnosis, so the focus typically shifts toward identifying the potential causal chromosomal or genetic mutation. Modern diagnostic pipelines rely on genetic and/or genomic testing, either targeting specific genes or regions (if a known syndrome or condition is suspected) or using whole-exome sequencing (WES), which is commonly employed to search for pathogenic variants affecting the coding part of the genome, sometimes supplemented by validation studies in the family and/or gene expression profiling [11]. As whole-genome sequencing (WGS) becomes more accessible and high-fidelity long-read sequencing technologies gain traction, an overall improvement in diagnostic rates is anticipated. These technologies enable more accurate detection of structural variants and genomic rearrangements, often missed by traditional short-read sequencing platforms [12].

However, in a substantial number of cases, these latest procedures still yield inconclusive results. Either no pathogenic variant is detected, the identified variants remain unannotated and unlinked to any known syndrome and deemed VUS [13], or no second hit is found in autosomal recessive disorders even though clinical suspicion exists. As a result, it is not possible to confidently correlate the observed phenotype with a definitive genetic cause. Further progress can be made if the VUS is found to be fully associated with the condition through functional assays such as protein modeling [14] and gene expression analysis. Consequently, while functional genomics-informed diagnostics have the potential to transform the rare disease diagnostic landscape, additional validation on individual approaches is still required. This common testing battery is illustrated in Figure 1.

One of the most promising avenues in functional genomic diagnostics of RDs is epigenetic analysis based on the detection of so-called episignatures (condition-specific position-based methylation differences in a group of cases compared with a reference cohort) and epimutations (sample-specific region-based methylation difference in a sample with respect to a reference cohort). DNA methylation is the addition of a methyl group to the 5-carbon of the cytosine ring, typically in the context of CpG dinucleotides, resulting in 5-methylcytosine [15,16]. Methylation is an absolute measure of the percentage of DNA sequences with a methyl group in a CpG site, making it more robust than other functional quantifications (transcripts or proteins) thanks to its comparability among all marks and samples without having to standardize across datasets. Methylation regulates gene expression, and if observed at an unexpected region (epimutation) may constitute a cause for an undiagnosed NDD. However, methylation at particular sites may also result from an underlying mutation, genetic alteration, or uncommon fetal exposure that might be the root of the RD, leading to global alterations in the epigenome [5,17,18]. These changes give rise to episignatures that enable the identification of differential epigenetic patterns between affected individuals and controls. These patterns have been associated with a wide array of RDs and NDDs [17,19,20] and can provide critical insights when conventional genetic testing falls short or is not easily accessible. Although DNA methylation can vary among different cells and tissue of an individual, the analysis of whole blood and/or other accessible tissue has proven useful for defining clinically relevant episignatures and epimutations.

Despite the significant potential of episignatures and epimutations to enhance RD diagnostics, several challenges still remain. These include: (1) assessing their true diagnostic value, (2) understanding their biological origins, and (3) standardizing methodologies for their identification and analysis. Additionally, the sensitive nature of genetic data presents privacy and security concerns, especially in clinical contexts involving undiagnosed cases, despite existing efforts within the scientific community to promote secure data sharing through data anonymization protocols and the creation of controlled repositories [21,22].

Experimental design for episignature detection in RDs has been established and well documented [23]. By contrast, analysis workflows are diverse and remain to be standardized. We therefore aimed to provide a comprehensive framework for episignature-based diagnosis of NDDs based on the recent literature to guide clinical practitioners and researchers in the field by addressing the following most current and salient issues. (1) How to streamline the development of new episignatures using established methodologies from the current literature? (2) How to process and analyze patient DNA methylation data to test episignatures using available models? (3) How to interpret episignature-based results with clinical relevance? And (4) what gaps still exist in the field, and how can these be overcome?

In this systematic review of the literature, we identify existing gaps in the field and discuss how addressing them will help in the implementation of epigenetic diagnostics for NDDs in clinical practice.

## 2. Materials and Methods

To describe current methods in episignature detection, we first reviewed studies that conducted epigenome-wide association studies (eWASs) in RDs and NDDs following the PRISMA guidelines [24]. Figure 2 illustrates the PRISMA flowchart with the included and excluded articles based on proposed filters. Specifically, we aimed to identify studies that measured the epigenetic profiles of patients diagnosed with RDs or NDDs, using blood samples, and that conducted differential DNA methylation analysis. Additionally, we included studies that developed classification algorithms based on episignatures and those that also studied epimutations. The article was registered on OSF and the registration code is fs9aj.

To ensure broad coverage, we included all reported types of DNA methylation profiling and methodologies. Only studies published between 2014 and 2025 were considered.

The articles for this review were sourced through the following strategies.

PubMed general search: Using the keywords “episignatures” OR “epimutation” filtered for publications between 2014 and 2025. Broad search parameters were applied due to the lack of standardized terminology in the field. The query was not restricted to RDs or NDDs to prevent missing out relevant articles. We retrieved 667 results. Titles and abstracts were screened to exclude case reports, general reviews without novel algorithmic or diagnostic contributions, and studies not directly related to NDDs or rare diseases. Filtering was guided by specific keywords such as “DNA methylation,” “signature,” and “profile.” Only 94 articles passed the proposed filters. In addition, we tracked new releases throughout 2025 using the same keywords in PubMed and added 7 extra studies.

Additional articles identified from the literature: 59 studies were retrieved from the literature focused on NDDs and RDs, as well as key large-scale studies citing references [25,26,27,28,29,30,31,32,33,34].

Overall, 160 articles were retained for full-text assessment. Particular attention was given to the Methods and Results sections to determine compliance with the following criteria: new episignatures or refinement of multiple episignatures with a new pipeline; blood-based to promote inter-study comparison; and exclusion of case studies and full-text availability. Ultimately, 108 studies that developed models or pipelines for one or more syndromes were included in the review.

## 3. Results

As of 2025, episignatures have been developed for well over 100 distinct genetic conditions (according to EpiSign v5 catalogue, Appendix A). In total, 106 different disorders were identified, and all studies retained from the literature are listed in Table 1, indicating for which disease an episignature was derived, the causal gene (if any), and the array type. The different paths for episignature testing are represented in Figure 3. Each step will be further explained in the sections below.

### 3.1. Testing Episignatures

#### 3.1.1. Testing Episignatures: External Resources

For users who want to test a given patient with the available episignatures, but do not have the bioinformatics background to apply the pipelines, multiple external resources are currently available for episignature testing: EpigenCentral, EpiSign, NSBEpi and MethaDory. The first two options offer comprehensive pipelines that process raw methylation array data (IDAT files) and return prediction scores indicating the likelihood of a sample being associated with a specific epigenetic disorder. The latter require some data processing.

EpigenCentral [25] is a publicly available web-based platform that enables users to analyze DNA methylation data and assess its association with episignature obtained by the developers. The platform currently supports autism spectrum disorder (*CHD8*/16p11.2 del), CHARGE syndrome (*CHD7*), Down syndrome (21 trisomy), Dup7 syndrome (7q11.23 dup), Kabuki syndrome (*KMT2D*), Nicolaides–Baraitser syndrome (*SMARCA2*), Sotos syndrome (*NSD1*), Weaver syndrome (*EZH2*), Williams–Beuren syndrome (7q11.23 del) (Appendix A). All steps are automatically performed, but are limited to available methods. After processing, the platform computes probabilistic scores indicating the likelihood of a sample belonging to a specific disease category using random forests, penalized logistic regression, or support vector machines.

For a complete overview of its functionalities, users are referred to the user guide found in their web application [137].

To our knowledge, only Awamleh et al. [138] has used the tool to generate an episignature anew from a Weaver syndrome cohort, yielding good predictions and case–control clustering when plotted using unsupervised clustering techniques. Further, the article thoroughly details which steps were followed in the platform to detect the episignature, explaining all needed files. Nevertheless, no external study has published results derived from the platform.

EpiSign [139] is a commercially available service that offers clinical-grade episignature diagnostics based on blood-derived DNA methylation profiling. Upon submission of a blood sample, the laboratory performs a methylation array analysis and applies their proprietary classification algorithm. Different pricing tiers are available depending on the scope of analysis requested (e.g., targeted panels vs. whole-platform screening). EpiSign is the main author in the episignatures field and possesses the most comprehensive algorithm developed so far, with more than 100 identifiable conditions on their version 5 platform (according to their web information, accessed on 1 October 2025). In Appendix A, a list is included of the offered episignatures.

The tool has been tested by the authors in multiple published studies [20,26,27,36,40], in which they provide excellent results of their predictors in most cases, including the resolution of cases with VUS. Nevertheless, since most of their data are not open-access due to ethical and corporative restrictions in genetic data sharing (as stated by the authors), external validation has not been extensively conducted.

Husson et al. [140] tested some of the episignatures from a past version of the EpiSign classifier [26], and other independent studies conducted in the same disease [49,61,126,133,141]: *ATRX*, ASD18 (*CHD8*), Cornelia de Lange syndrome (*NIPLB*), CHARGE (*CHD7*), Kabuki (*KMT2D*), Claes–Jensen (*KDM5C*), Rubinstein-Taybi (*CREBBP*), Sotos (*NSD1*), Tatton-Brown–Rahman (*DNMT3A*), and Wiedemann–Steiner (*KMT2A*). The authors describe three types of episignatures: robust, which allow a clear identification of cases and the top PC explains most of the detected variance (the case for ATRX, Sotos, TBRS and Kabuki derived episignatures); unstable, which offers high sensitivity though intermediate profiles, thus not allowing perfect distinction of all cases or certain VUS (the case for Cornelia de Lange, CHARGE, ASD18, Claes–Jensen- and Wiedemann–Steiner-derived episignatures); and weak, which did not work as expected, and the authors advise against their usage (the case for Rubinstein–Taybi and a second AUTS18-derived episignature).

The NSBEpi (nanopore sequencing-based episignature detection) pipeline was developed to test long-read sequencing episignature discovery [142]. The authors demonstrated the feasibility of training classification models using only one positive case against other disease cases and controls averaged by their median methylation value at episignature-defined sites. Their classification models are available on their GitHub repository (JorisVermeeschLab/NSBEpi, version: 21 January 2025), trained over the episignatures developed by Aref-Eshghi et al. [26]. Although developed for sequencing, the tool can also be adapted for array data [34].

MethaDory is a recent open-access shiny application (in beta testing) which incorporates publicly available episignatures and classification models even for those syndromes without open-access data available [34] (GitHub repository: f-ferraro/MethaDory, version: 19 October 2025). The authors designed a method to create synthetic profiles with which they were able to train classification models based on past episignatures using the median-beta approach reported by Geysens et al. [142]. Since the software is in its first stages, not all data are readily available, but the authors offer their models upon request (Appendix A).

Although full validation of some existing episignatures is still pending, results are promising, with many allowing highly specific and sensitive identification of cases, even when VUS are involved.

#### 3.1.2. Testing Episignatures: Public Episignatures

In addition to the previously mentioned tools, an episignature can be tested on a set of samples using two common strategies in methylation analysis:

Unsupervised clustering: Techniques such as principal component analysis (PCA), multidimensional scaling (MDS), or hierarchical clustering can be applied to methylation data at disease-relevant CpG sites, using the sites defining the episignature. This step is performed in all studies developing or testing episignatures to assess how well cases and controls are separated. While this heuristic approach does not yield a direct probability score, it can highlight whether an unknown case clusters with known affected individuals, thereby supporting or guiding a diagnostic hypothesis. Nevertheless, it requires confirmed cases of the suspected disease for comparison.

Machine learning-based prediction: When a trained classification model is available, individual samples can be scored to estimate the likelihood of belonging to a specific disease class. This approach does not require comparison with other cases or controls. Currently, EpigenCentral and NSBEpi offer its models open-access (and MethaDory but upon request), so this technique is limited to these tools. This approach is exemplified in the study conducted by Hildonen et al. [143], who used the CpGs describing the *KMT2D* episignature [26] (for Kabuki syndrome) to test a set of undiagnosed carriers of *KMT2D* patients, demonstrating the utility of robust episignatures for diagnostic purposes. Lastly, Niceta et al. [144] tested another population of putative Kabuki patients with *KMT2D* VUS and mosaicism using three different public episignatures [26,61,75] after training independent classification models with confirmed Kabuki patients, which proved fruitful. These studies highlight the potential of episignatures for elusive cases and variant reclassification.

### 3.2. Episignature Development

If data are available (either private data or open-access from databases such as Gene Expression Omnibus (GEO), ArrayExpress or the European Genome-Phenome archive (EGA)), researchers and clinicians can derive their own episignatures. The process consists of three main computational steps: raw data preprocessing (a general methylation processing step), differential methylation analysis (a general methylation analysis step with specific considerations for RDs and NDDs), and model training (specific to NDDs). While the third step is not essential for episignature identification, it allows the assignment of confidence values to individual samples based on the identified significant sites.

A complete description of the pipeline is shown in Figure 4, and the main approaches used in each study are summarized in Appendix A. The following sections provide a detailed explanation of each step.

#### 3.2.1. Episignature Development: Data Formats

Methylation can be routinely measured using both methylation arrays and genome sequencing [145]. Methylation arrays screen specific predefined CpG sites clustered mostly in regions of interest (i.e., gene promoters, gene bodies, regulatory regions, etc.), with newer versions covering up to 930,000 different sites. Genome sequencing in the form of bisulfite sequencing or Oxford Nanopore/PacBio long reads offers more comprehensive methylation detection, though requires more computational capabilities.

To date, most episignatures have been developed using Illumina methylation arrays, specifically with versions 450 K or EPICv1, with a few exceptions where bisulfite sequencing via Illumina short read was employed [76,92,107]. However, the most recent episignatures already use the newest EPICv2 version.

Illumina arrays store raw data in binary .”idat” (intensity data) files, with each subject represented by two files—green and red channels—corresponding to array chemistry. Raw reads, on the other hand, are stored as .”fastq” files, and a series of steps are required to call methylation data per CpG before any downstream analysis.

The following preprocessing options are not specific to RD settings, but represent a common approach for all methylation analysis projects. Specific changes might be introduced due to the characteristics of studies using RDs, which will be specified in each study (Appendix A). In addition, Table 2 includes a summary of all packages and tools in the reviewed articles to conduct the analysis.

#### 3.2.2. Episignature Development: DNA Methylation Array Processing

For studies screening patients using arrays, raw data require several preprocessing steps. Data analysis is predominantly conducted in R with open-access Bioconductor packages used in the literature—minfi [146], RnBeads [152,153], ChAMP [147,148], lumi [149], meffil [151], wateRmelon [157], and SeSAMe [150]—in addition to Illumina Genome Studio software. The choice of package often depends on the research group’s experience since each one implements similar algorithms, although minfi has been the most frequently used.

.idat files are uploaded into R at the probe level, utilizing both color channels. Background correction is applied to distinguish true signal from background noise [177], typically based on negative control probes present in the array. Normalization is then performed to ensure comparability across samples and arrays within an experiment [178]. Several normalization methods exist for Illumina arrays, including SWAN [179], quantile normalization [180], noob normalization [177], functional normalization [181], Illumina Genome Studio default normalization and beta mixture quantile dilation (BMIQ) [182]. For further discussion about the topic, see [101,183], who tested the different processing methods using an RD dataset and in general samples, respectively.

For 450 K arrays, MethylAid [156] was developed as a Shiny application that uses minfi-associated methods for interactive array processing and methylation analysis. The package was used to perform basic quality control forf two fetal alcohol spectrum disorder-focused studies [80,81].

For RDs, no consensus was found in the selected articles about processing methodologies, though Illumina, functional, and noob normalizations were the most used and considered the most reliable ones [101], albeit the final selection may depend on cohort size.

Probe-level intensities are then merged at the CpG level, generating methylated and unmethylated signals, or an aggregated cumulative signal. CpG methylated/unmethylated signals can be transformed into quantitative aggregated values that summarize the methylation status of each cytosine: beta values and M-values. For an extensive description of the parameters, we refer to [184,185]. In short, each value represents:Beta_CpGi = max(CpG_i_meth_intensity, 0)/(max(CpG_i_meth_intensity, 0) + max(CpG_i_unmeth_intensity, 0) + alpha)(1)

Beta values (1) range from 0 to 1, with values > 0.8 and <0.2 indicating high and low methylation, respectively, but are not optimal for formal statistical testing due to their non-normal distribution [184].M_CpGi = log2(max(CpG_i_meth_intensity, 0)/max(CpG_i_unmeth_intensity, 0))(2)

M-values (2) range from −∞ (fully unmethylated) to +∞ (fully methylated) and are better suited for statistical modeling because they approximate a normal distribution, although less intuitive to interpret [184].

To ensure data reliability, several QC metrics are applied at both probe and sample levels.

At the probe level, common QC metrics consider: (1) Probes near single-nucleotide variants (SNVs) with minor allele frequencies (MAFs) > 1%. (2) Cross-reactive probes. (3) Probes with raw beta values of 0 or 1 in >0.25% of samples. (4) Non-CpG probes. (5) Sex chromosome probes (X and Y) are often excluded. (6) Probes with detection *p*-value > 0.05 or 0.01.

Cross-reactive probes and those near SNVs are commonly removed using standardized probe lists available in R packages, such as minfi, ChAMP, SeSAMe, or maxprobes, which offer automated filtering functions using standard lists of CpGs.

At the sample level: (1) Methylation values should follow a bimodal distribution, typically centered on ~0.2 and ~0.8 (beta values), reflecting the tendency of CpGs to be either almost fully unmethylated or methylated, respectively. (2) Samples with a high percentage of unreliable probes (e.g., with detection *p*-value >0.05 or >0.01).

The ultimate choice of QC thresholds and filters can vary depending on the specific objectives of the analysis.

#### 3.2.3. Episignature Development: Multi-Cohort Studies

Merging datasets is often required to achieve statistical power for episignature detection in RD studies. When working with merged datasets, and as a normal QC step, batch inspection and correction is critical. Tools like ComBat (from the sva package [158] or its implementation in ChAMP) are used to correct for technical and biological variation across batches/datasets, while preserving key phenotypic differences (i.e., ensuring that correction does not obscure real biological signals).

Several studies used ComBat to correct batch effects (i.e., technical sources of variation) prior to episignature detection, though in all instances technical issues were identified through a PCA or MDS. Different batch sources and explanatory covariates were considered when modeling batches. For instance, Kagami et al. [98] only identified technical batches (array ID) and not clinical covariates while analyzing two imprinting syndromes (Temple and Kagami–Ogata syndromes); the same setting was repeated by Carmel et al. when studying the association of 22q11.2 deletion and schizophrenia [42]; a study that derived an episignature for autism [33] included sex as a potential confounding factor, even though X/Y probes were discarded; in a Williams syndrome study [134], HNRNPU-related neurodevelopmental disorder [68] and Aicardi–Goutières [43] identified both sex and age as potential batch effects. Lastly, two studies deriving the RNU4-2 episignature [123] and PURA-related neurodevelopmental disorders [118] also used ComBat, but did not specify which covariates of interest were accounted for to maintain biological differences.

Before applying any batch correction, researchers should first evaluate whether such correction is necessary. This can be accomplished using dimensionality-reduction and clustering techniques such as PCA, MDS, or hierarchical clustering to detect potential batch effects or sample outliers. If a batch effect is identified and corrected (array ID, processing site if multiple laboratories were involved, or processing date), then the result must subsequently be evaluated again to ensure that it was applied appropriately. In particular, investigators should verify that the adjustment eliminated the batch and did not introduce artifacts, such as inflated differences or excessive shrinkage that undermines confidence in the corrected methylation values. If batch correction proves unsuitable or introduces undesirable distortions, an alternative approach is to work with the uncorrected data while statistically modeling the batch as a covariate and/or incorporating surrogate variables to account for latent sources of heterogeneity [158,186].

#### 3.2.4. Episignature Development: Methylation Screening Using Sequencing Methods

Bisulfite sequencing is another DNA methylation screening usable for episignatures; however, it is not a common choice, with only three reported studies using it [76,92,107]. Raw bisulfite data consist of .fastq files storing the sequence read and the quality of each base (i.e., Phred scores). Three basic steps are required before downstream methylation analysis is conducted: preprocessing, genome alignment, and methylation calling.

All three studies used trim-galore [187] to preprocess raw reads, eliminating adaptors and applying QC filters based on phred scores. Bisulfite sequencing requires specific alignment algorithms due to reduced read complexity while increasing mismatch mappings with the reference genome [188]. In all cases, alignment was conducted using bismark [189], which allows the detection of methylated sites by adding the option “bismark methylation extractor.” The output is stored in .”bismarkCov” files, indicating the genomic position and the ratio of methylated vs. total reads, akin to beta values. .”bismarkCov” files can be read and loaded into R using RnBeads for downstream analysis. Quality control metrics include exclusion of CpGs found in X and Y chromosomes and a minimum 10× coverage.

Long reads (PacBio and Oxford Nanopore), despite screening all genomic and methylomic variants at the same time, have not been systematically used for episignature detection yet [190]. This might be due to their added analysis complexity, requiring higher storage capacity and computer capabilities, and increased cost, limiting their full implementation. Interestingly, a proof-of-concept study demonstrating that nanopore sequencing enables episignature detection for up to 13 different disorders has recently been published, illustrating the possibility to use episignatures developed using arrays in sequencing profiles [142], although variant detection was reported to be not as accurate as using short-read sequencing. Moreover, epimutations have been studied using PacBio-HiFi in another RD cohort [190], proving the possibility to detect allele-specific hypermethylation in regulatory regions (i.e., epimutations).

#### 3.2.5. Episignature Development: Differentially Methylated Position (DMP) Detection

Episignatures are sets of CpGs that consistently exhibit differential methylation between conditions, making them valuable predictive biomarkers. Typically, episignatures are identified by fitting a linear model across the epigenome and selecting CpGs that surpass predefined thresholds of statistical significance and methylation differences.

The limma R package is the most widely used tool, modeling M values per CpG while incorporating relevant covariates to control for confounding factors and derive a list of significant DMPs. Some commonly used R packages (minfi, meffil, RnBeads, ChAMP) offer built-in functions in their pipelines, though these are rarely used. Most studies account for at least immune cell composition, while others also include variables such as age, sex, and known batches to minimize confounding. These covariates represent key sources of biological and technical variation that, if not considered, can lead to spurious associations. However, the selection of appropriate covariates is not always straightforward. The epigenetics community widely acknowledges blood heterogeneity as a significant source of biological variation [191,192,193]. This is particularly relevant for episignature development, as cohort variability in age [193], inflammation, and immune-deficiency-related diseases [194,195,196] can significantly alter blood composition and consequently the epigenetic landscape.

A clear example of this challenge is exemplified during the development of the Nicolaides–Baraitser syndrome episignature [113], where high correlations among blood cells were detected. The authors concluded that only monocytes should be considered a confounding variable in DMP analysis.

Among the available deconvolution methods, the algorithm developed by Houseman et al. [197] remains the most commonly used, although there is no clear consensus on which method is best for rare-disease patients. For instance, some studies used the Houseman algorithm [61,113], while most recent manuscripts [48,130] used FlowSorted.Blood.EPIC [161]. A recent benchmarking of current algorithms and their performance [198] concluded that EpiDISH [162] outperforms other available methods, although none of the reviewed studies used it.

Disagreement also extends to immune cells types that should be included in the models. For instance, Butcher et al. [61] and Velasco et al. [94] did not include cell heterogeneity for HARGE and Kabuki studies or for multiple disorders caused by alterations in the epigenetic machinery, respectively. Chater-Diehl et al. [113] only adjusted for monocyte proportion for a Nicolaides–Baraitser study, while Siu et al. and Choufani et al. [49,130] considered six common cell types (B cells, natural killer, CD4+ T cells, CD8+ T cells, monocytes and neutrophils) using FlowSorted.Blood.EPIC and Houseman’s algorithm for *EZH2* variants and *CHD8*/16p11.2del studies, respectively. To address uncertainty in cell composition modeling, it has been suggested that the top surrogate variables [199] be included, although only two of the reviewed studies used surrogate variables to account for heterogeneity, in addition to blood cells [52,102].

Those studies that screened methylation by bisulfite sequencing [76,92,107] also derived episignatures fitting a linear Nicolaides–Baraitser model, though through a limma implementation in RnBeads.

Despite M values being more statistically suitable for DMP detection, several studies used beta values instead of M values for their analysis and obtained positive results [32,37,38,57,60,66,81,87,88,90,106,110,134,200].

To summarize the methodologies in latest releases, methods have already diverged: both RNU4-2 episignatures [122,123] used meffil and functional normalization for data processing. A new version of the fetal alcohol spectrum disorder episignature [82] and one derived for *PTBP1* carriers [117] used SeSAMe without specifying which normalization method was employed. Lastly, two studies used minfi to process *MORC2* variants in multiple diseases [60] coupled with quantile normalization and in Sifrim–Hitz–Weiss patients (*CHD4* variants) [125], though without specifying the method. Except in Nava et al.’s study [122], who used the limma wrapper from meffil, all other episignatures were derived using the standard limma linear model with commonly chosen covariates: chronological or epigenetic age, sex, immune cell proportions, and/or array ID.

#### 3.2.6. Episignature Development: Other Approaches to Linear Models for Episignatures

Beyond linear models, various alternative strategies have been applied to identify significant DMPs, though not used in recent studies. Prickett et al. [32] compared methylated and unmethylated intensities between cases and controls to find hyper- and hypomethylated sites in Silver–Russell patients. Choufani et al. [126] used a nonparametric Mann–Whitney U test in Sotos syndrome patients adjusting for multiple comparisons using the Bonferroni method. Kagami et al. [98] applied a rule-based method, selecting CpGs with at least a 0.05 beta-value difference and more than three standard deviations between groups to define significant sites in Kagami–Ogata syndrome.

The Partek Genomics Suite provided by Illumina https://help.partek.illumina.com/partek-genomics-suite (accessed on 3 October 2025) was used by four studies [44,59,84,86] to perform DMP analysis. These studies used ANOVA tests with differing thresholds to detect differentially methylated regions (DMRs), applying different criteria to study alpha thalassemia, adult-onset autosomal dominant cerebellar ataxia, floating-harbor syndrome, and fragile X syndrome, respectively.

Finally, Siu et al. [49] developed two pairs of episignatures for ASD patients with 16p11.2 deletions and *CHD8* variants. Their strategy involved initial filtering of the most variable sites across batches using Qlucore Omics Explorer https://qlucore.com/ (accessed on 3 October 2025), followed by the intersection of significant results using limma and Mann–Whitney U tests to develop the final episignature. The most recent study that did not use a linear model built in R was conducted by Oussalah et al. [78] in 2022, who used Golden Helix’s SNP and Variation Suite https://www.goldenhelix.com/products/SNP-Variation/ (accessed on 3 October 2025) and MedCalc to analyze a cohort with a vitamin B12 metabolism disorder. These platforms allowed PCA to assess sample distribution and outlier detection, followed by DMP identification using pairwise *t*-tests.

#### 3.2.7. Episignature Development: DMRs and Epimutations

A different strategy involves identifying differentially methylated regions (DMRs). Biologically speaking, CpGs clustered within a DMR are more likely to regulate nearby genes collectively, making regional methylation patterns more informative than specific CpG sites [201]. Two primary well-established methods for identifying DMRs in the reviewed literature are bumphunter [165], implemented in minfi, and DMRcate [163]. Both can also be implemented directly using the ChAMP package and are based on linear models on multiple CpG sites in a genomic locus. Bumphunter fits a linear model to identify CpGs differentially methylated associated with the trait of interest, and then merges together into significant regions using permutation tests. DMRcate again fits a linear model to find differentially methylated CpGs, though it then scales the CpGs using a Gaussian kernel, assigning higher weights to spatially close CpGs.

Some studies designed custom algorithms to test more restrictive approaches. In 2019, Garg et al. [50] applied a 1 kb sliding window approach, defining a DMR as a region where at least three probes met stringent methylation difference thresholds (≥99.9 or ≤0.01 percentile of control distribution, with absolute differences of ≥0.15) for an ASD and schizophrenia cohort. A more recent approach conducted by LaFlamme et al. [67] in 2024 used a similar approach, but with different thresholds (≥99.25th or ≤0.75th percentile), and accounted for sex-specific X chromosome methylation patterns rather than excluding them altogether in pediatric epilepsies and patients with *CHD2* variants.

Besides searching for consensus DMRs, other authors focused on detecting epimutations per sample, offering a patient-specific analysis. In 2018, Barbosa et al. [129] designed a custom sliding window to compare each patient’s reference profile, differentiating for autosomal and X CpGs, with a reference cohort panel to find hypermethylation and hypomethylation events. Their cohort consisted of a mix of congenital disorders, often without any known diagnosis. For this reason, common DMR and DMP approaches could not be conducted due to their high variability.

In 2023, Lee et al. [92] screened a cohort of patients with pathogenic variants in *HNRNPU* using NGS and detected differentially methylated regions by clustering physically close significant DMPs using a two-sided Welch test or by a linear model implemented in limma. Only CpGs functionally similar (i.e., all found in the same CpG island) were combined.

Despite their potential, DMR-based approaches are not the primary focus of episignature research, as episignatures are typically defined at the CpG-site level. CpGs within a significant region tend to be highly correlated, rendering them redundant if used as separate features in a classification model. However, recent discussions [101] have explored the feasibility of using DMRs for episignature testing, particularly as a means to ensure compatibility with future array versions, since they showed DMR-derived episignatures are less affected by missing data.

Regardless of the approach, once DMPs or DMRs are identified, they can be used to investigate gene dysregulation, cluster cases and controls based on methylation profiles, and provide insights into genotype-phenotype associations. In addition, an extra step is usually taken to filter which CpGs to include to represent the true episignature from all statistically significant ones (i.e., with an adjusted *p*-value < 0.05). The different approaches will be described in depth below in the section Episignature Development: Classification Model Pre-Filtering.

#### 3.2.8. Episignature Development: Annotation and Enrichment

Another key step is the annotation of significant loci to determine any possible association with the patients’ phenotype. This step allows the identification of whether the episignature is biologically explanatory.

The most straightforward method to annotate DMPs is using the annotation files provided by Illumina and easily accessible through specific Bioconductor packages (IlluminaHumanMethylation450kanno.ilmn12.hg19, IlluminaHumanMethylationEPICanno.ilm10b4.hg19, and IlluminaHumanMethylationEPICv2anno.20a1.hg38), which contain the mapped gene (if any) and the genomic context (i.e., if the CpG is part of a CpG island), among others. In the case of DMRs, the most interesting aspect is finding which genes fall within each region and if they encompass any regulatory element of interest. These can be annotated using packages such as annotatr [169] or LOLA [170].

Enrichment analysis can be conducted to find CpGs associated with known pathways. When analyzing methylation data, enrichment should take into account the positional correlation of each CpG and multiple CpG mapping the same gene such as missMethyl [171] or methylGSA [173], though common enrichment techniques have also been employed using GSEA [174].

Another approach was applied by Kimura et al. [134], who used the WGCNA R package [164] to detect co-methylation correlation patterns among different genes. While the idea might be similar to detect DMRs, their approach tested correlations among independent CpGs throughout the genome without the constraint of being physically close, finding gene networks that might be involved in causal pathways. Kimura et al. tested a cohort of Williams syndrome patients, finding genes associated with ASD and Alzheimer’s disease, among other neuropsychiatric phenotypes related to Williams syndrome.

#### 3.2.9. Episignature Development: Classification Model Introduction

An important objective in episignature detection is the classification of new individuals. Different machine learning (ML) methods can be applied to methylation-based diagnosis of RD. The goal of ML methods is to build an algorithm capable of predicting whether a given epigenetic pattern originates from a patient affected by a specific disease. In other words, it aims to assign a confidence score indicating the likelihood that a sample belongs to a patient with a given disease versus an unaffected individual.

#### 3.2.10. Episignature Development: Classification Model Pre-Filtering

A first key step is the filtering of informative features derived from any of the aforementioned methods. Filtering of candidate DMPs is crucial to prevent overfitting, particularly in cases where low-effect CpGs with minimal predictive power are retained [184]. There is large variability in filtering strategies, reflecting either disease-specific epigenetic alterations that require tailored approaches or the iterative tuning needed to obtain a usable set of CpGs that best discriminates cases and controls given the limited sample size. This variability hinders episignature comparisons and model standardization, since filtering decisions are rarely reported.

Hence, common approaches in this case include applying thresholds to both the adjusted *p*-value (typically ranging from 0.1 to 0.01 under Benjamini–Hochberg correction) and the absolute methylation difference (Δβ), which generally needs to exceed 0.05 (5% methylation difference). The exact thresholds vary, as exemplified in the study conducted by Aref-Eshghi et al. [36], in which 14 different RDs were tested. Sotos syndrome had the most stringent threshold (Δβ > 0.25), while BAFopathies, Cornelia de Lange syndrome, and CHARGE syndrome had the less restrictive ones (Δβ > 0.05).

Many different studies refined this approach by ranking features based on the product of the Δβ and the −log(adjusted *p*-value) and selecting the top-ranked probes [26,29,35,40,41,51,53,56,58,60,62,64,66,68,69,72,74,79,82,88,89,96,97,103,109,110,111,112,114,117,120,121,123,125,133,135]. Additional filtering was applied based on probe–probe correlation (typically using Spearman’s correlation with thresholds below 0.7 and 0.9) and individual probe performance metrics such as AUC to assess disease discrimination, keeping a given number of CpGs, which again changed from study to study or within a given study. For instance, in a Menke–Hennekam study [109] of patients with *CREBBP* or *EP300* mutations, the authors tried to identify three different episignatures, which depended on in which domain the variant was found (ZZ, TAZ2, or ID4). For each variant, the authors decided independent thresholds of significance and the number of probes to keep. Only patients with ID4 mutations resulted in a clinically significant episignature, though mild alterations were found for ZZ- and TAZ2-variant carriers.

A different implementation was used by Oexle et al. [75] to account for probe redundancy. A minimum redundancy–maximum relevance (mRMR) ensemble algorithm was used, implemented in the mRMRe R package [167], to leverage importance and correlation between features, allowing a reduction in the gross episignature length while maintaining the signature. In a multi-class study, Levy et al. [38], after filtering by Δβ and adjusted *p*-value, merged all significant CpGs from all cases and trained a multi-class random forest to select the most informative features for each class.

In other studies, minimal filtering was performed beyond statistical significance and methylation difference, especially when feature selection was integrated within the model itself, allowing the algorithm to choose the most informative features during training [85,101,106,119].

#### 3.2.11. Episignature Development: Classification Model Design

While training a machine learning (ML) model, several aspects must be considered to ensure reliable results.

Validation strategy: If additional samples are not available for testing (ideally external cohorts), some cases and controls should be set aside for model testing and evaluation (usually 25% of the cohort, usually fewer than five samples; Table 3). Nevertheless, if the cohort is small, withholding some data for validation may reduce the model’s robustness.

Variant-level specificity: Should the model be trained on all individuals collectively or should it focus on specific variants? A given disease might be caused by myriad mutations and/or causal genes. For instance, Rubinstein–Taybi syndrome, mainly caused by mutations in *CREBBP* or *EP300* [202], has gene-specific episignatures rather than syndrome-specific., or even at gene-domain level, as reported by Haghshenas et al. [109] in Menke–Hennekam patients, caused by variants in the same two genes.

Class imbalance: large differences in group sizes (e.g., many more controls than cases) can bias the model towards predicting the majority class. For example, if controls outnumber cases 10:1, the model might favor “control” predictions by default, since most predictions would be correct. Several strategies exist to control class imbalance: upsampling, creating synthetic data by subsampling with replacement features from the minor class; downsampling, by subsampling the majority class and training the model iteratively on different subsets of the same size as the minority class; the synthetic minority oversampling technique (SMOTE), a mixed approach that downsamples the majority class and creates new synthetic data for the minority one [203]; or random oversampling example (ROSE), which generates synthetic data of the minority class by adding noise to existing data points [204].

Cross-validation (CV) and hyperparameter tuning: Model performance depends heavily on the choice of hyperparameters (i.e., parameters set before training the ML model, which will influence the final model quality and generalizability [205]), which should be optimized using cross-validation (CV) loops (Figure 5). Evaluation metrics such as precision, AUC, or F1 score guide feature selection [206,207] (Table 3). A common strategy is X-fold CV (Figure 5 exemplifies fivefold CV with three repetitions) or leave-one-out CV (holding out a single sample per fold).

**Table 3 biomedicines-13-03043-t003:** Model evaluation metrics used during CV and to assess model performance. In estimation column: TPs (true positives), TNs (true negatives), FPs (false positives), FNs (false negatives), log L($\hat{\theta }$) (log-likelihood).

Metric	Definition	Estimation	Ref.
Recall or sensitivity	Proportion of real positives correctly identified.	(TPs)/(TPs + FNs)	[206]
Specificity	Proportion of real negatives correctly identified.	(TNs)/(TNs + FPs)	[206]
Accuracy	Proportion of correctly classified instances.	(TPs + TNs)/(TPs + TNs + FPs + FNs)	[206]
Precision	Proportion of predicted positives that are actually positive.	(TPs)/(TPs + FPs)	[206]
AUC	Area under the ROC curve; represents trade-off between sensitivity and specificity.	Computed from ROC curve	[207]
F1 score	Harmonic mean of precision and sensitivity.	(2 × precision × sensitivity)/(precision + sensitivity)	[206]
Deviance	Comparison between trained model and perfect model.	−2 log L$\hat{\theta }$	[208]
Cohen’s kappa	Comparison between model predictions (Pr(a)) and random guessing (Pr(e)).	(Pr(a) − Pr(e))/(1 − Pr(e))	[209]

Most recent models have been developed using the caret R package [210], which implements various classification algorithms under a single tool, providing integrated solutions for managing validation strategy, class imbalance, and cross-validation. Specifically, caret offers encapsulation and common function call for all commonly used algorithms: support vector machines (SVMs), random forests (RFs), penalized logistic regression (PLR), elastic net, and coarse approximation linear function. However, ML models are available from specific software packages for each classification model type.

The most comprehensive predictor so far, EpiSign, has been developed using SVMs (with linear kernel). SVMs are a class of supervised machine learning algorithms used for classification and regression. Their core principle is to find the optimal hyperplane that best separates the data into distinct classes. In two dimensions, this hyperplane is simply a line, but in higher dimensions, it corresponds to a more complex topology. The optimal hyperplane is defined as the one that maximizes the margin, or distance between the nearest data points of each class, known as support vectors. In Refs. [211,212], SVM model training is implemented in e1071 and caret R packages.

SVMs are trained based on a kernel function, indicating how the hyperplane should be built: Linear kernel is suitable when classes are linearly separable. The main hyperparameter is the cost, a regularization parameter that prevents overfitting by balancing between low training error and large margin. Radial basis kernel maps data into nonlinear dimensional space. Besides the cost, it is also defined with the sigma hyperparameter, defining the influence each training sample has, leveraging overall model complexity.

Random forests are ensemble learning models in which many classifiers (trees) are generated and the final decision accomplished by aggregating their results. RFs are based on growing several independent and identically distributed decision trees based on different bootstrap subsets of the training data, which offer discriminatory cutoffs to decide an outcome based on the most informative features selected from an initial random subset of them [213,214,215]. RFs are based on two hyperparameters: number of variables per tree and number of trees in the forest. Multiple strategies to train RFs are seen in the literature. Ciolfi et al. [119] trained a model in caret for Rahman syndrome patients with C-terminal frameshift mutations, and Kimura et al. [33] used the randomForest R package [215] to train a classifier for adults with high-functioning ASD. In addition, other Python-based tools were also used for RF training: Giuili et al. [101] and Turinsky et al. [25] used scikit-learn [216] and GenPipes [176], respectively.

Other popular ML methods in methylation analysis, but less exploited in the context of rare diseases are penalized logistic regression (PLR) and elastic net. Penalized methods prevent model overfitting thanks to adding shrinkage parameters to less informative features and controlling the overall importance of each feature using a regularization parameter λ [217,218]. Two methods exist: Lasso (or L_1_ penalization) shrinks coefficients of non-informative variables to zero, providing effective variable selection when inter-feature correlation is low. Ridge (or L_2_ penalization) retains all features with non-zero coefficients, performing better with highly correlated features and low overall noise.

Elastic net leverages both methods according to an α parameter (ridge: α = 0; lasso: α = 1; in between, elastic net). In the literature, logistic regression approaches have only been implemented using scikit-learn and GenPipes [25,101], despite the availability of packages in R such as caret and glmnet [218].

Another method seen in the literature is the coarse approximation linear function using the CALF R package, a greedy algorithm that builds a classification metric by adding each feature one at a time based on Student *t*-test *p*-values until a newly added feature does not improve the prediction capabilities anymore. CALF has only been applied once, in a high-functioning ASD cohort [33], accompanied by ROC as evaluation metric.

In addition to formal ML models, a simpler approach was used in the past consisting of a median-based decision algorithm [48,49,113,130]. Classification was accomplished by estimating the median difference of episignature CpGs between cases and controls.

Overall, the studies reported strong predictive performance, regardless of the analytical method (most did not report balancing strategy or which evaluation metric was used), disease type, or screening approach. Most validations were internal, typically performed by predicting the status of individuals left out before model training. Despite this lack of external validation, it is important to emphasize that such validation is essential for establishing the true generalizability of a model. When discovery and validation samples are processed together, observed case–control differences may reflect not only genuine biological variation but also technical or batch effects. This can substantially diminish discriminatory performance in an independent cohort, as was the case in some episignatures discussed in earlier sections. Despite this, Ferraro et al.’s approach to training independent models based on synthetic data using known episignatures proved that most of them could be validated (though not using the same model as the original article) [34].

## 4. Discussion

We have presented the current state of the art regarding episignatures and epimutation analysis for RD diagnosis after a thorough systematic revision of the literature. Episignature-based diagnosis is likely to become a reference tool in the functional evaluation of RDs. Nevertheless, several challenges must be addressed to standardize their usage in clinical practice.

(1) **Limited case availability and data sharing:** Clinical application limited by the scarcity of available cases, hindering external validations. There is a notable lack of open-access data in public repositories. In RD research, sharing genetic data is even more difficult due to the uniqueness of patient’s profiles. Even though early studies had published the raw data in open-access repositories, data are not commonly shared anymore, as stated by the authors, owing to ethical or legal concerns. In recent years, data sharing has been hindered due to concerns of patient re-identification [219,220], which could partially explain the reduced number of public releases, but since many studies are conducted by private initiatives that offers episignatures as a service, the underlying decision could be influenced by economic reasons.

Regardless of data-sharing issues, episignature inference requires as many cases as possible to reliably capture disease-specific epigenomic patterns. New approaches have been proposed to comply with data protection policies [21,221], enabling controlled access for research centers to refine existing episignatures or validate new ones. Repositories with controlled access such as EGA could be used so public institutions can decide which authors can access the data to conduct specific analysis. DataSHIELD is a framework that allows research centers and hospitals to securely connect either database through a federated, non-disclosive approach, ensuring data confidentiality [222,223]. DataSHIELD, or other federated data networks, are particularly suitable for RD research [224,225], since they could offer the possibility of using more data to call episignatures or train classification models without actually sharing or disclosing them. Briefly, federated data networks are structured databases under certain data harmonization rules, which are fully controlled by the research center in question, offering non-disclosive access to other users. This way, data governance and patient confidentiality is not compromised [226], though the data are accessible to foster new developments. Regardless, recently developed tools (NSBEpi [142] and MethaDory [34]) demonstrate the possibility of working with a reduced number of cases, offering a good alternative to create custom classification models from known episignatures if no more cases are available.

(2) **Array version updates:** Microarray versions are regularly updated, causing older arrays to become deprecated. Certain CpGs may no longer be measured, limiting the applicability of episignatures developed on legacy platforms like 450K and EPIC due to missing CpGs [101] if not imputed. The EPICv2 array, for instance, uses updated annotations, complicating direct translation [227]. Two different strategies exist to address this issue:DMRs represented by a single methylation statistic (mean, median, or quantile metric) [101,201]. Under this approach, all CpGs found in a region take part in the metric, regardless of which specific CpGs are taken into account.Sequencing via genome bisulfite sequencing or third-generation “five-base” callers. The latter offers promising results, since third-generation sequencing technologies could offer the detection of all available genetic variability, from intronic/exonic variants to whole- genome methylation [190].

However, there exists cross-platform probe mapping to promote compatibility. Some Bioconductor R packages (SeSAMe and MethylCallR [155]) offer this conversion, which is even more essential for EPICv2, since multiple probes can map the same CpG.

(3) **Methodological diversity:** Heterogeneous analysis pipelines hinder diagnostic standardization. This variability prevents objective comparisons and limits the development of unified workflows, especially since original cohorts are often inaccessible for peer review. Confounding factors in DMP calling are particularly challenging. Blood cell deconvolution methods were trained in the general population or in cancer, hence using them in RDs with inflammatory phenotypes or global immune defects may misestimate cell proportions. Likewise, post-DMP CpG filtering lacks standardized guidelines. Although many approaches are functionally similar and differ only in their software implementation, the overall lack of consistency can make pipelines appear tailored to individual datasets rather than to the disease itself. This in turn increases the risk of model overfitting and reduces reproducibility in external cohorts.

(4) **Tissue specificity:** Methylation is tissue-specific. Blood-derived episignatures may not reflect changes in affected tissue, such as the brain in NDDs. For that reason, the obtained CpGs might not have any biological meaning, but may work as a biomarker likewise. Alternative tissue (e.g., cerebrospinal fluid or saliva) could be explored, as they may be more biologically relevant or easier to obtain. Mosaicism may also yield inconsistent episignatures depending on the affected tissue and mosaicism extent.

(5) **Longitudinal stability:** To date, episignatures have been assessed cross-sectionally at a single time point, without follow-up studies evaluating the stability of the profiles. However, methylation studies outside episignature research have tested this stability. Komaki et al. [228] screened the epigenome at 24 time points throughout three months in two control adults and did not detect any major longitudinal differences. In a lupus study, Coit et al. [229] screened 54 patients during 4 years, and again the authors detected a stable profile. However, in a third study using a control children cohort [230], variation was detected throughout the first 5 years of life. Hence, since many RDs and NDDs are studied in children, extra focus should be placed on the area on the grounds of developing meaningful episignatures regardless of age.

(6) **Chromosomal sex stratification:** Sex is often ignored as a confounding variable. Methylome differences between sexes are well documented during puberty [231] and in the general population [232], potentially affecting episignatures. In addition, many studies discard XY probes, disregarding the epigenetic effect on the gonosomes. X probes can be informative, either through non-sex-specific deviations or disease-related interactions. For instance, Radio et al. [35] studied patients with 1p36 deletion syndrome spanning the *SPEN* gene, which was reported as a modulator in X chromosome inactivation [233], and found 122 significant CpGs in the X chromosome, but this only worked in females.

(7) **Lack of integration among developed episignatures for the same disease:** Independent episignatures for the same disease are rarely compared or standardized [140]. Niceta et al. [144] found that three different episignatures for Kabuki syndrome could differentiate most cases and classify them correctly. While each episignature was based on different CpGs, they correctly classified patients with *KMT2D* variants. A different approach was taken by Davide et al. [28], who tried to integrate multiple episignatures by conducting a meta-analysis using open-access datasets of ASD patients. The pooled significant sites could thus be used as a consensus episignature for the disease.

(8) **Clinical utility and VUS interpretation:** Episignatures are valuable for functional diagnosis, in unresolved cases, or for clinical confirmation. Episignatures can add crucial information on VUS, particularly when other tests are costly or unavailable. Notably, Ciolfi et al. [119] developed an RF classifier for Rahman syndrome. The authors ran the classifier against known and unresolved cases and identified a patient with a positive episignature, later confirmed via exome sequencing. The same approach has worked for other syndromes [143,144] to confirm cases with VUS.

(9) **Study of imprinting disorders:** Some RDs can be directly caused by abnormal methylation of imprinted genes and regions [234]. The molecular diagnosis of these syndromes might be better targeted at the analysis of these causal regions rather than using whole episignatures, as discussed by Hara-Isono et al., in which no common episignature could be found in Silver–Russell patients with different molecular etiologies [235]. Another study focused on four different imprinting disorders (Angelman, Prader–Willi, Beckwith–Wiedemann and Silver–Russell syndrome) also had the same approach of analyzing specific regions, which demonstrated the utility of this approach as a diagnostic tool [236]. Variant-specific episignatures may be achievable [235], but focusing on epimutation events at targeted loci could yield more consistent diagnoses.

Finally, this review has some limitations that should be pointed out. Although we aimed to capture the full breadth of episignature research, our search strategy may have missed relevant studies, particularly because terminology varies and some reports do not explicitly use the term “episignature.” The field is largely driven by a relatively small number of groups, introducing the possibility of publication bias and limiting the visibility of alternative methodologies. Moreover, most original datasets and trained classifiers are not publicly accessible, preventing direct, head-to-head performance comparisons across approaches and constraining us to the descriptions provided by the original authors. Negative results are rarely published, so the number of disorders without a confirmed episignature is likely underreported. Until such non-confirmatory findings become more widely available, some conditions may continue to be tested repeatedly even if a robust episignature does not exist.

## 5. Conclusions

Episignatures represent a promising advancement for improving the diagnosis of RDs and NDDs. They offer a powerful framework for developing new clinical tools that can support and refine diagnostic decision-making. To date, an expanding number of disease-specific episignatures have been identified, highlighting their diagnostic potential across diverse syndromes. Although formal studies in routine clinical settings are still limited, the available evidence suggests that incorporating validated episignatures and epimutations into standard diagnostic pipelines could substantially increase diagnostic yield. Nevertheless, further efforts are required to ensure methodological robustness, promote standardization across analytical workflows, and facilitate their effective integration into clinical practice.

## Figures and Tables

**Figure 1 biomedicines-13-03043-f001:**
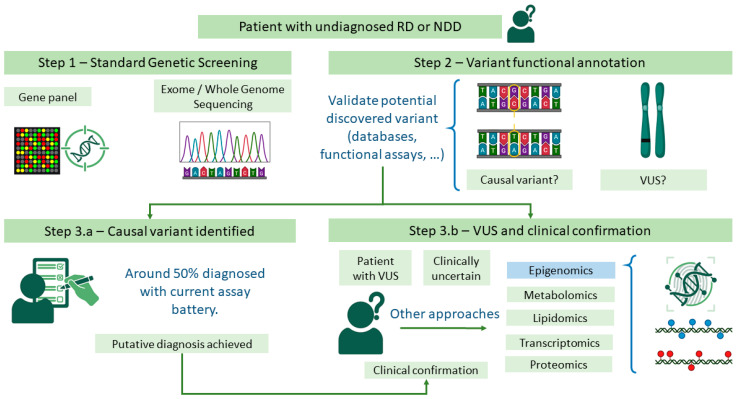
Global approach in genetic testing for a rare disease (RD). First, either targeted (for suspected cases) or global methods are used to detect variants. Afterwards, the variant must be annotated according to the literature. If the variant can be considered causal and matches the clinical diagnosis, the patient might be diagnosed. If either no variant or a variant of uncertain significance (VUS) is found or the identified variant does not match the clinical phenotype, additional techniques are required to reach a diagnosis. However, additional testing can also be conducted to validate genetic variants. Icons downloaded from BioRender.org. There is no copyright issue.

**Figure 2 biomedicines-13-03043-f002:**
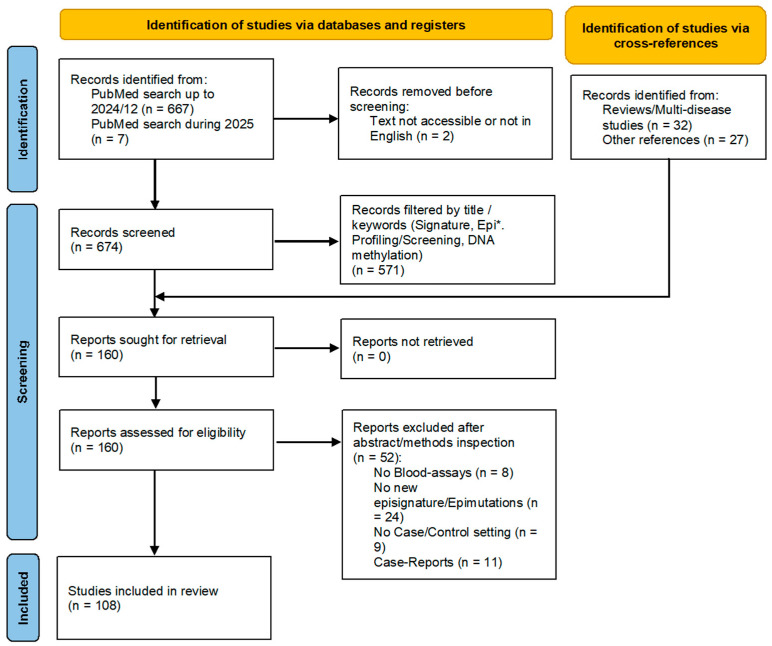
PRISMA flowchart for article selection and bibliographic filters considered during the first stage of article review. There is no copyright issue.

**Figure 3 biomedicines-13-03043-f003:**
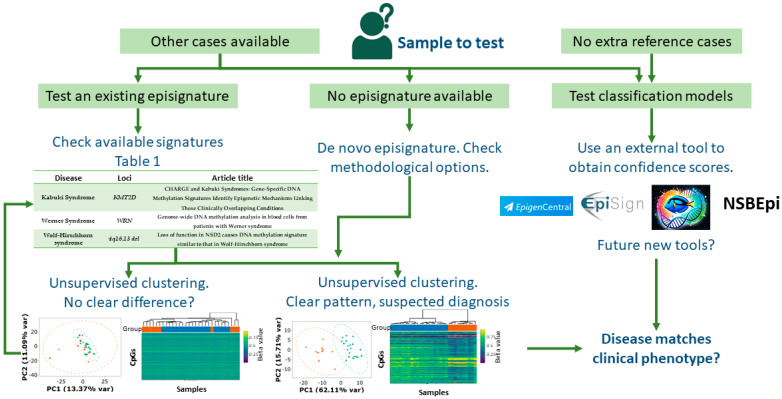
Proposed pipeline for episignature detection. Two main paths exist depending on whether other methylation profiles are available for comparison. If not, an external facility is required. If so, the user can either develop their own episignature or test any open-access model, searching for the disease of interest by consulting the studies summarized in this review. In the principal component analysis (PCA) and hierarchical clustering colors indicate case (orange) and control (green/blue) samples. Icons downloaded from BioRender.org. There is no copyright issue.

**Figure 4 biomedicines-13-03043-f004:**
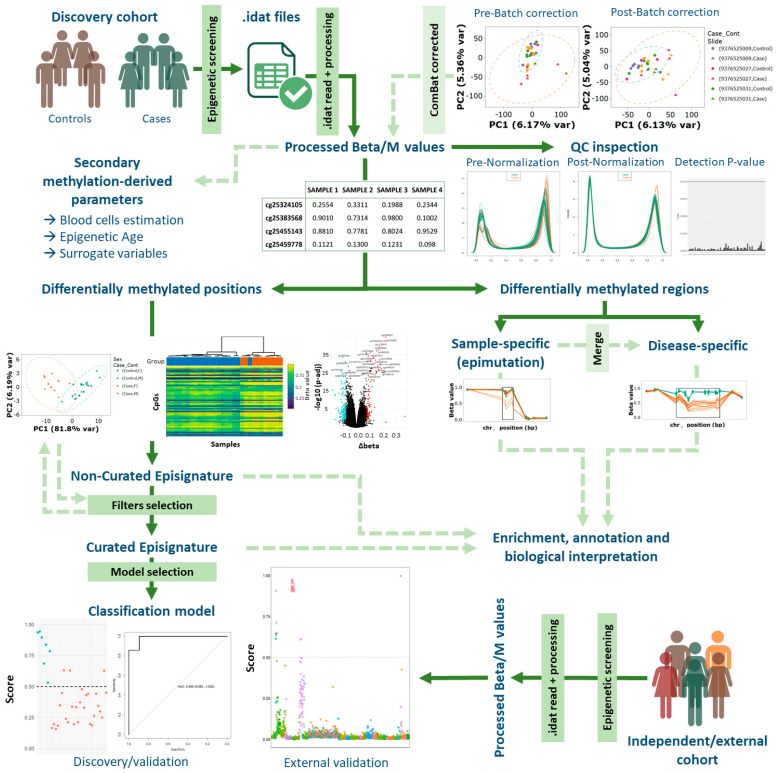
Full episignature detection pipeline using methylation arrays. Continuous dark green lines and boxes indicate steps conducted in most studies or considered key. Dotted light-green lines and boxes indicate processes that sometimes are not conducted, but were considered important to keep in mind. In the PCA, hierarchical clustering, epimutation plot, and discovery/validation plot, colors indicate cases (orange) and controls (green/blue). In the external validation plot, colors represent different diseases to illustrate validation using cases of the studied episignature (positive controls) and controls plus other cases that should not respond to the episignature (negative controls). Icons downloaded from BioRender.org. There is no copyright issue.

**Figure 5 biomedicines-13-03043-f005:**
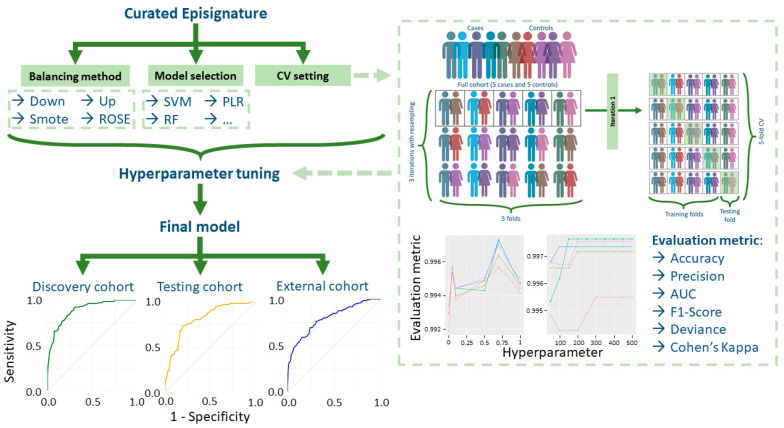
Summary of key steps in classification model training. CpGs have already been filtered and curated. Different balancing methods, ML model types, and validation settings are available. Before the final model is trained, the most optimal hyperparameters should be found, with internal CV rounds selecting an evaluation metric. The colors in evaluation metric plots indicate different combination of model-specific hyperparameters. The final model should be tested to assess its performance (in the example, with ROC curves). Icons downloaded from BioRender.org. There is no copyright issue.

**Table 1 biomedicines-13-03043-t001:** List of included articles with the studied disease, altered gene or variant, and screening method. “*” indicates a locus erroneously annotated in the article. “---” indicates no specific gene. Genes marked /*/ found no episignature even though other genes causing the disease yielded significant results.

Disease	Loci	Variant	Array	Article Title	Ref.
1p36 deletion syndrome	1p36, *SPEN*	Deletion in females	EPIC	*SPEN* haploinsufficiency causes a neurodevelopmental disorder overlapping proximal 1p36 deletion syndrome with an episignature of X chromosomes in females	[35]
6q24–q25 deletion syndrome	6q24–q25	Deletion	450 k, EPIC	Diagnostic utility of genome-wide DNA methylation testing in genetically unsolved individuals with suspected hereditary conditions	[36]
7q11.23 duplication syndrome	7q11.23	Duplication	450 k	Symmetrical dose-dependent DNA-methylation profiles in children with deletion or duplication of 7q11.23	[37]
	7q11.23	Duplication	450 k, EPIC	Diagnostic utility of genome-wide DNA methylation testing in genetically unsolved individuals with suspected hereditary conditions	[36]
	7q11.23	Duplication	450 k, EPIC	Evaluation of DNA methylation episignatures for diagnosis and phenotype correlations in 42 Mendelian neurodevelopmental disorders	[26]
	7q11.23	Duplication	450 k, EPIC	Functional correlation of genome-wide DNA methylation profiles in genetic neurodevelopmental disorders	[38]
9q34.3 microduplication syndrome	9q34.3	Microduplication	EPIC	Refining the 9q34.3 microduplication syndrome reveals mild neurodevelopmental features associated with a distinct global DNA methylation profile	[39]
16p11.2 deletion syndrome	16p11.2	Deletion	450 k, EPIC	Novel diagnostic DNA methylation episignatures expand and refine the epigenetic landscapes of Mendelian disorders	[40]
	16p11.2	Deletion	450 k, EPIC	Functional correlation of genome-wide DNA methylation profiles in genetic neurodevelopmental disorders	[38]
22q11.2 deletion syndrome (velocardiofacial syndrome)	22q11.2	Deletion	EPIC	Identification of a DNA Methylation Episignature in the 22q11.2 Deletion Syndrome	[41]
	22q11.2	Deletion	EPIC	Differential methylation of imprinting genes and MHC locus in 22q11.2 deletion syndrome-related schizophrenia spectrum disorders	[42]
	22q11.2	Deletion	450 k, EPIC	Novel diagnostic DNA methylation episignatures expand and refine the epigenetic landscapes of Mendelian disorders	[40]
	22q11.2	Deletion	450 k, EPIC	Functional correlation of genome-wide DNA methylation profiles in genetic neurodevelopmental disorders	[38]
Aicardi–Goutières syndrome	*RNASEH2B*		EPIC	Altered DNA methylation and gene expression predict disease severity in patients with Aicardi–Goutières syndrome	[43]
Alpha-thalassemia/impaired intellectual development syndrome, X-linked	*ATRX*		450 k	Identification of epigenetic signature associated with alpha thalassemia/mental retardation X-linked syndrome	[44]
	*ATRX*		450 k	Genomic DNA methylation signatures enable concurrent diagnosis and clinical genetic variant classification in neurodevelopmental syndromes	[45]
	*ATRX*		450 k, EPIC	Diagnostic utility of genome-wide DNA methylation testing in genetically unsolved individuals with suspected hereditary conditions	[36]
	*ATRX*		450 k, EPIC	Evaluation of DNA methylation episignatures for diagnosis and phenotype correlations in 42 Mendelian neurodevelopmental disorders	[26]
	*ATRX*		450 k, EPIC	Novel diagnostic DNA methylation episignatures expand and refine the epigenetic landscapes of Mendelian disorders	[40]
	*ATRX*		450 k, EPIC	Functional correlation of genome-wide DNA methylation profiles in genetic neurodevelopmental disorders	[38]
Arboleda–Tham syndrome	*KAT6A*		450 k, EPIC	Novel diagnostic DNA methylation episignatures expand and refine the epigenetic landscapes of Mendelian disorders	[40]
	*KAT6A*		450 k, EPIC	Functional correlation of genome-wide DNA methylation profiles in genetic neurodevelopmental disorders	[38]
*ARID2*-related disorder	*ARID2*		EPIC	*ARID2*-related disorder: further delineation of the clinical phenotype of 27 novel individuals and description of an epigenetic signature	[46]
Attention deficit/hyperactivity disorder (ADHD)			450 k	DNA methylation epi-signature and biological age in attention deficit hyperactivity disorder patients	[47]
Au–Kline syndrome	*HNRNPK*	Missense and loss of function	EPIC	An *HNRNPK*-specific DNA methylation signature makes sense of missense variants and expands the phenotypic spectrum of Au-Kline syndrome	[48]
Autism spectrum disorder (ASD)	16p11.2 del		450 k, EPIC	Functional DNA methylation signatures for autism spectrum disorder genomic risk loci: 16p11.2 deletions and *CHD8* variants	[49]
	*CHD8*				
			450 k	An epigenetic biomarker for adult high-functioning autism spectrum disorder	[33]
			450 k, EPIC	Epigenetics of autism spectrum disorders: a multi-level analysis combining epi-signature, age acceleration, epigenetic drift and rare epivariations using public datasets	[28]
			450 k	Screening for rare epigenetic variations in autism and schizophrenia	[50]
	*CHD8*		450 k, EPIC	Evaluation of DNA methylation episignatures for diagnosis and phenotype correlations in 42 Mendelian neurodevelopmental disorders	[26]
	*CHD8*		450 k, EPIC	Novel diagnostic DNA methylation episignatures expand and refine the epigenetic landscapes of Mendelian disorders	[40]
			450 k, EPIC	Functional correlation of genome-wide DNA methylation profiles in genetic neurodevelopmental disorders	[38]
Autosomal dominant intellectual developmental disorder—65 (MRD65)	*KDM4B*		450 k, EPIC	Novel diagnostic DNA methylation episignatures expand and refine the epigenetic landscapes of Mendelian disorders	[40]
	*KDM4B*		450 k, EPIC	Functional correlation of genome-wide DNA methylation profiles in genetic neurodevelopmental disorders	[38]
BAFopathy nonsyndromic	*ARID1A*, *ARID1B*	Duplications	EPIC	Microduplications of ARID1A and ARID1B cause a novel clinical and epigenetic distinct BAFopathy	[51]
Beck–Fahrner syndrome	*TET3*		450 k, EPIC	Novel diagnostic DNA methylation episignatures expand and refine the epigenetic landscapes of Mendelian disorders	[40]
	*TET3*		450 k, EPIC	Functional correlation of genome-wide DNA methylation profiles in genetic neurodevelopmental disorders	[38]
Berardinelli-Seip Congenital Lipodystrophy type 2 (CGL2)	*BSCL2*		EPIC	Accelerated epigenetic aging and DNA methylation alterations in Berardinelli–Seip congenital lipodystrophy	[52]
*BCL11B*-related disease (BCL11B-RD)	*BCL11B*		EPIC	Clinico-biological refinement of *BCL11B*-related disorder and identification of an episignature: a series of 20 unreported individuals	[53]
Beck–Fahrner syndrome	*TET3*		EPIC	Deficiency of *TET3* leads to a genome-wide DNA hypermethylation episignature in human whole blood		[54]
Blepharophimosis intellectual disability syndrome (BIS)	*SMARCA2*	Exons 8 and 9	EPIC	De novo *SMARCA2* variants clustered outside the helicase domain cause a new recognizable syndrome with intellectual disability and blepharophimosis distinct from Nicolaides-Baraitser syndrome	[55]
	*SMARCA2*		EPIC	Blepharophimosis with intellectual disability and Helsmoortel-Van der Aa syndrome share episignature and phenotype	[56]
	*SMARCA2*		450 k, EPIC	Novel diagnostic DNA methylation episignatures expand and refine the epigenetic landscapes of Mendelian disorders	[40]
	*SMARCA2*		450 k, EPIC	Functional correlation of genome-wide DNA methylation profiles in genetic neurodevelopmental disorders	[38]
Börjeson–Forssman–Lehmann syndrome (BFLS)	*PHF6*		450 k, EPIC	Evaluation of DNA methylation episignatures for diagnosis and phenotype correlations in 42 Mendelian neurodevelopmental disorders	[26]
	*PHF6*		450 k, EPIC	Novel diagnostic DNA methylation episignatures expand and refine the epigenetic landscapes of Mendelian disorders	[40]
	*PHF6*		450 k, EPIC	Functional correlation of genome-wide DNA methylation profiles in genetic neurodevelopmental disorders	[38]
Bohring–Opitz syndrome (BOS)	*ASXL*		EPIC	DNA methylation signature associated with Bohring-Opitz syndrome: a new tool for functional classification of variants in *ASXL* genes	[57]
*CDK13*-related disorder (CDK13-RD)	*CDK13*		EPIC	*CDK13*-related disorder: report of a series of 18 previously unpublished individuals and description of an epigenetic signature	[58]
Cerebellar ataxia, deafness and narcolepsy, autosomal dominant (ADCADN)	*DNMT1*		450 k	Identification of a methylation profile for *DNMT1*-associated autosomal dominant cerebellar ataxia, deafness, and narcolepsy	[59]
	*DNMT1*		450 k	Genomic DNA methylation signatures enable concurrent diagnosis and clinical genetic variant classification in neurodevelopmental syndromes	[45]
	*DNMT1*		450 k, EPIC	Diagnostic utility of genome-wide DNA methylation testing in genetically unsolved individuals with suspected hereditary conditions	[36]
	*DNMT1*		450 k, EPIC	Evaluation of DNA methylation episignatures for diagnosis and phenotype correlations in 42 Mendelian neurodevelopmental disorders	[26]
	*DNMT1*		450 k, EPIC	Novel diagnostic DNA methylation episignatures expand and refine the epigenetic landscapes of Mendelian disorders	[40]
	*DNMT1*		450 k, EPIC	Functional correlation of genome-wide DNA methylation profiles in genetic neurodevelopmental disorders	[38]
Charcot–Marie–Tooth disease type 2Z (CMT2Z)	*MORC2*		EPIC, EPICv2	Pleiotropic effects of *MORC2* derive from its epigenetic signature	[60]
CHARGE syndrome	*CHD7*		450 k	CHARGE and Kabuki syndromes: gene-specific DNA methylation signatures identify epigenetic mechanisms linking these clinically overlapping conditions	[61]
	*CHD7*		EPIC	Genomic DNA methylation signatures enable concurrent diagnosis and clinical genetic variant classification in neurodevelopmental syndromes	[45]
	*CHD7*		450 k, EPIC	Diagnostic utility of genome-wide DNA methylation testing in genetically unsolved individuals with suspected hereditary conditions	[36]
	*CHD7*		450 k, EPIC	Evaluation of DNA methylation episignatures for diagnosis and phenotype correlations in 42 Mendelian neurodevelopmental disorders	[26]
	*CHD7*		450 k, EPIC	Novel diagnostic DNA methylation episignatures expand and refine the epigenetic landscapes of Mendelian disorders	[40]
	*CHD7*		450 k, EPIC	Functional correlation of genome-wide DNA methylation profiles in genetic neurodevelopmental disorders	[38]
Chung–Jansen syndrome	*PHIP*		EPIC	The detection of a strong episignature for Chung–Jansen syndrome, partially overlapping with Börjeson–Forssman–Lehmann and White–Kernohan syndromes	[62]
Claes–Jensen syndrome	*KDM5C*		450 k	Peripheral blood epi-signature of Claes-Jensen syndrome enables sensitive and specific identification of patients and healthy carriers with pathogenic mutations in *KDM5C*	[63]
	*KDM5C*		450 k	Genomic DNA methylation signatures enable concurrent diagnosis and clinical genetic variant classification in neurodevelopmental syndromes	[45]
	*KDM5C*		450 k, EPIC	Diagnostic utility of genome-wide DNA methylation testing in genetically unsolved individuals with suspected hereditary conditions	[36]
	*KDM5C*		450 k, EPIC	Evaluation of DNA methylation episignatures for diagnosis and phenotype correlations in 42 Mendelian neurodevelopmental disorders	[26]
	*KDM5C*		450 k, EPIC	Functional correlation of genome-wide DNA methylation profiles in genetic neurodevelopmental disorders	[38]
Clark–Baraitser syndrome	*TRIP12*		EPIC	Episignature mapping of *TRIP12* provides functional insight into Clark-Baraitser syndrome	[64]
Coffin–Lowry syndrome	*RSK2*		450 k	Genomic DNA methylation signatures enable concurrent diagnosis and clinical genetic variant classification in neurodevelopmental syndromes **(Not significant)**	[45]
Coffin–Siris syndrome	*ARID1B*, *SMARCB1*		450, EPIC	Diagnostic utility of genome-wide DNA methylation testing in genetically unsolved individuals with suspected hereditary conditions	[36]
	*ARID1B*		450 k, EPIC	BAFopathies’ DNA methylation epi-signatures demonstrate diagnostic utility and functional continuum of Coffin–Siris and Nicolaides–Baraitser syndromes	[65]
	*SMARCB1*		450 k, EPIC		
	*SOX11*		EPIC	*SOX11* variants cause a neurodevelopmental disorder with infrequent ocular malformations and hypogonadotropic hypogonadism and with distinct DNA methylation profile	[66]
	*ARID1B*, *SMARCB1*		450 k	Genomic DNA methylation signatures enable concurrent diagnosis and clinical genetic variant classification in neurodevelopmental syndromes **(Not significant)**	[45]
	*ARID1A*, *ARID1B*, *SMARCB1*, *SMARCA4*		450 k, EPIC	Evaluation of DNA methylation episignatures for diagnosis and phenotype correlations in 42 Mendelian neurodevelopmental disorders	[26]
	*ARID1A*, *ARID1B*, *SMARCB1*, *SMARCA4*, *SOX11*		450 k, EPIC	Novel diagnostic DNA methylation episignatures expand and refine the epigenetic landscapes of Mendelian disorders	[40]
	*ARID1A*, *ARID1B*, *SMARCB1*, *SMARCA4*, *SOX11*		450 k, EPIC	Functional correlation of genome-wide DNA methylation profiles in genetic neurodevelopmental disorders	[38]
	*ARID1A*, *ARID1B*	*c6200*			
	*SMARCA4*	*c2650*			
Cohen–Gibson syndrome	*EED*		450 k, EPIC	Novel diagnostic DNA methylation episignatures expand and refine the epigenetic landscapes of Mendelian disorders	[40]
	*EED*		450 k, EPIC	Functional correlation of genome-wide DNA methylation profiles in genetic neurodevelopmental disorders	[38]
Cockayne syndrome (CS)	*MORC2*		EPIC, EPICv2	Pleiotropic effects of *MORC2* derive from its epigenetic signature	[60]
Cornelia de Lange syndrome	*NIPBL*, *RAD21*, *SMC1A*, *SMC3*		450 k, EPIC	Diagnostic utility of genome-wide DNA methylation testing in genetically unsolved individuals with suspected hereditary conditions	[36]
	*NIPBL*, *RAD21*, *SMC3*, *SMC1A*		450 k, EPIC	Evaluation of DNA methylation episignatures for diagnosis and phenotype correlations in 42 Mendelian neurodevelopmental disorders **(Not significant)**	[26]
	*HDAC8*				
	*NIPBL*, *RAD21*, *SMC1A*, *SMC3*, *HDAC8*		450 k, EPIC	Novel diagnostic DNA methylation episignatures expand and refine the epigenetic landscapes of Mendelian disorders	[40]
	*NIPBL*, *SMC1A*, *SMC3*, *RAD21*		450 k, EPIC	Functional correlation of genome-wide DNA methylation profiles in genetic neurodevelopmental disorders	[38]
Developmental and epileptic encephalopathy	*CHD2, ---*		450 k, EPIC	Diagnostic utility of DNA methylation analysis in genetically unsolved pediatric epilepsies and *CHD2* episignature refinement	[67]
	*HNRNPU*		EPIC	DNA methylation episignature and comparative epigenomic profiling of *HNRNPU*-related neurodevelopmental disorder	[68]
Developmental delay with gastrointestinal, cardiovascular, genitourinary, and skeletal abnormalities syndrome (DEGCAGS)	*ZNF699*		EPIC	Epigenomic and phenotypic characterization of DEGCAGS syndrome	[69]
DOT1L-associated syndrome	*DOT1L*	Increase of function	EPIC	Rare de novo gain-of-function missense variants in *DOT1L* are associated with developmental delay and congenital anomalies	[70]
Down syndrome	+chr21	Trisomy	450 k	Identification of a DNA methylation signature in blood cells from persons with Down syndrome	[71]
	+chr21	Trisomy	450 k, EPIC	Diagnostic utility of genome-wide DNA methylation testing in genetically unsolved individuals with suspected hereditary conditions	[36]
	+chr21	Trisomy	450 k, EPIC	Evaluation of DNA methylation episignatures for diagnosis and phenotype correlations in 42 Mendelian neurodevelopmental disorders	[26]
	+chr21	Trisomy	450 k, EPIC	Novel diagnostic DNA methylation episignatures expand and refine the epigenetic landscapes of Mendelian disorders	[40]
	+chr21	Trisomy	450 k, EPIC	Functional correlation of genome-wide DNA methylation profiles in genetic neurodevelopmental disorders	[38]
Duchenne muscular dystrophy	*DMD*		EPIC	The discovery of the DNA methylation episignature for Duchenne muscular dystrophy	[72]
*DYRK1A* intellectual disability	*DYRK1A*	Loss of function	EPIC	Integrative approach to interpret *DYRK1A* variants, leading to a frequent neurodevelopmental disorder	[73]
Dystonia 28, childhood onset	*KMT2B*		EPIC	Childhood-onset dystonia-causing *KMT2B* variants result in a distinctive genomic hypermethylation profile	[74]
	*KMT2B*		EPIC	Episignature analysis of moderate effects and mosaics	[75]
	*KMT2B*		NGS	Comparison of methylation episignatures in *KMT2B*- and *KMT2D*-related human disorders	[76]
	*KMT2B*		EPIC	Blood DNA methylation provides an accurate biomarker of *KMT2B*-related dystonia and predicts onset	[77]
	*KMT2B*		450 k, EPIC	Novel diagnostic DNA methylation episignatures expand and refine the epigenetic landscapes of Mendelian disorders	[40]
	*KMT2B*		450 k, EPIC	Functional correlation of genome-wide DNA methylation profiles in genetic neurodevelopmental disorders	[38]
epi-cblC disease	*PRDX1*, *MMACHC*	515-1G > T (*MMACHC* epimutation)	EPIC	*PRDX1* gene-related epi-cblC disease is a common type of inborn error of cobalamin metabolism with mono- or bi-allelic *MMACHC* epimutations	[30]
	*TESK2*, *MMACHC*	Epimutation	450 k	Epimutations in both the *TESK2* and *MMACHC* promoters in the Epi-cblC inherited disorder of intracellular metabolism of vitamin B12	[78]
Epileptic encephalopathy, childhood onset	*CHD2*		450 k, EPIC	Evaluation of DNA methylation episignatures for diagnosis and phenotype correlations in 42 Mendelian neurodevelopmental disorders	[26]
	*CHD2*		450 k, EPIC	Novel diagnostic DNA methylation episignatures expand and refine the epigenetic landscapes of Mendelian disorders	[40]
	*CHD2*		450 k, EPIC	Functional correlation of genome-wide DNA methylation profiles in genetic neurodevelopmental disorders	[38]
Fanconi anemia	*FANCA*		EPIC	Identification of a robust DNA methylation signature for Fanconi anemia	[79]
Fetal alcohol spectrum disorder (FASD)			450 k	DNA methylation abundantly associates with fetal alcohol spectrum disorder and its subphenotypes	[80]
			450 k	Expression quantitative trait methylation analysis identifies whole blood molecular footprint in fetal alcohol spectrum disorder (FASD)	[81]
			EPICv2	Discovery of a DNA methylation episignature as a molecular biomarker for fetal alcohol syndrome	[82]
Fetal valproate syndrome (SVF)			EPIC	Discovery of DNA methylation signature in the peripheral blood of individuals with history of antenatal exposure to valproic acid	[83]
FG syndrome (also Opitz–Kaveggia syndrome)	*MED12*		450 k, EPIC	Evaluation of DNA methylation episignatures for diagnosis and phenotype correlations in 42 Mendelian neurodevelopmental disorders **(Not significant)**	[26]
	*MED12*		450 k, EPIC	Novel diagnostic DNA methylation episignatures expand and refine the epigenetic landscapes of Mendelian disorders	[40]
Floating-harbor syndrome	*SRCAP*		450 k	The defining DNA methylation signature of floating-harbor syndrome	[84]
	*SRCAP*		EPIC	Truncating *SRCAP* variants outside the floating-harbor syndrome locus cause a distinct neurodevelopmental disorder with a specific DNA methylation signature	[85]
	*SRCAP*		450 k	Genomic DNA methylation signatures enable concurrent diagnosis and clinical genetic variant classification in neurodevelopmental syndromes	[45]
	*SRCAP*		450 k, EPIC	Diagnostic utility of genome-wide DNA methylation testing in genetically unsolved individuals with suspected hereditary conditions	[36]
	*SRCAP*		450 k, EPIC	Evaluation of DNA methylation episignatures for diagnosis and phenotype correlations in 42 Mendelian neurodevelopmental disorders	[26]
	*SRCAP*		450 k, EPIC	Novel diagnostic DNA methylation episignatures expand and refine the epigenetic landscapes of Mendelian disorders	[40]
	*SRCAP*		450 k, EPIC	Functional correlation of genome-wide DNA methylation profiles in genetic neurodevelopmental disorders	[38]
Fragile X	*FMR1*	GCC repetitions and deletion	450 k	Clinical validation of fragile X syndrome screening by DNA methylation array	[86]
Frontotemporal dementia (FTD)	17q21.31		450 k	An epigenetic signature in peripheral blood associated with the haplotype on 17q21.31, a risk factor for neurodegenerative tauopathy	[87]
Gabriele–de Vries syndrome (GADEVS)	*YY1*		EPIC	DNA methylation episignature in Gabriele-de Vries syndrome	[88]
	*YY1*		450 k, EPIC	Novel diagnostic DNA methylation episignatures expand and refine the epigenetic landscapes of Mendelian disorders	[40]
	*YY1*		450 k, EPIC	Functional correlation of genome-wide DNA methylation profiles in genetic neurodevelopmental disorders	[38]
Genitopatellar syndrome (GTPTS)	*KAT6B*		450 k	Genomic DNA methylation signatures enable concurrent diagnosis and clinical genetic variant classification in neurodevelopmental syndromes	[45]
	KAT6B		450 k, EPIC	Diagnostic utility of genome-wide DNA methylation testing in genetically unsolved individuals with suspected hereditary conditions	[36]
	*KAT6B*		450 k, EPIC	Evaluation of DNA methylation episignatures for diagnosis and phenotype correlations in 42 Mendelian neurodevelopmental disorders	[26]
	*KAT6B*		450 k, EPIC	Novel diagnostic DNA methylation episignatures expand and refine the epigenetic landscapes of Mendelian disorders	[40]
	*KAT6B*		450 k, EPIC	Functional correlation of genome-wide DNA methylation profiles in genetic neurodevelopmental disorders	[38]
Glass syndrome	*SATB2*		450 k, EPIC	Evaluation of DNA methylation episignatures for diagnosis and phenotype correlations in 42 Mendelian neurodevelopmental disorders **(Not significant)**	[26]
	*SATB2*		450 k, EPIC	Novel diagnostic DNA methylation episignatures expand and refine the epigenetic landscapes of Mendelian disorders	[40]
Hao–Fountain syndrome	*USP7*		EPIC	DNA methylation episignature, extension of the clinical features, and comparative epigenomic profiling of Hao-Fountain syndrome caused by variants in *USP7*	[89]
Helsmoortel–Van der Aa Syndrome (HVDAS)	*ADNP*	Class I (outside 2000–2340 bp)	EPIC	Episignatures sStratifying Helsmoortel-Van der Aa syndrome show modest correlation with phenotype	[90]
		Class II (inside 2156 and 2317)			
	*ADNP*		EPIC	Blepharophimosis with intellectual disability and Helsmoortel-Van Der Aa Syndrome share episignature and phenotype	[56]
	*ADNP*	Central Terminal	EPIC	Gene domain-specific DNA methylation episignatures highlight distinct molecular entities of *ADNP* syndrome	[91]
	*ADNP*	Central Terminal	450 k, EPIC	Diagnostic utility of genome-wide DNA methylation testing in genetically unsolved individuals with suspected hereditary conditions	[36]
	*ADNP*	Central Terminal	EPIC	Evaluation of DNA methylation episignatures for diagnosis and phenotype correlations in 42 Mendelian neurodevelopmental disorders	[26]
	*ADNP*	Central Terminal	450 k, EPIC	Novel diagnostic DNA methylation episignatures expand and refine the epigenetic landscapes of Mendelian disorders	[40]
	*ADNP*	Central Terminal	450 k, EPIC	Functional correlation of genome-wide DNA methylation profiles in genetic neurodevelopmental disorders	[38]
*HNRNPU*-related syndrome	*HNRNPU*		NGS	Germline pathogenic variants in *HNRNPU* are associated with alterations in blood methylome	[92]
Hunter–McAlpine syndrome	*NSD1*, 5q35 *	Duplication	450 k, EPIC	Evaluation of DNA methylation episignatures for diagnosis and phenotype correlations in 42 Mendelian neurodevelopmental disorders	[26]
	*NSD1*, 5q35	q terminal duplication	450 k, EPIC	Novel diagnostic DNA methylation episignatures expand and refine the epigenetic landscapes of Mendelian disorders	[40]
	*NSD1*, 5q35	q terminal duplication	450 k, EPIC	Functional correlation of genome-wide DNA methylation profiles in genetic neurodevelopmental disorders	[38]
Immunodeficiency with centromeric instability and facial anomalies (ICF) syndrome	*DNMT3B*, *ZBTB24*, *CDCA7*, *HELLS*		450 k	Comparative methylome analysis of ICF patients identifies heterochromatin loci that require *ZBTB24*, *CDCA7* and *HELLS* for their methylated state	[93]
	*DNMT3B*, *ZBTB24*, *CDCA7*, *HELLS*		450 k	Interplay between histone and DNA methylation seen through comparative methylomes in rare Mendelian disorders	[94]
	*DNMT3B*, *CDCA7*, *ZBTB24*, *HELLS*		450 k, EPIC	Evaluation of DNA methylation episignatures for diagnosis and phenotype correlations in 42 Mendelian neurodevelopmental disorders	[26]
	*DNMT3B*, *CDCA7*, *ZBTB24*, *HELLS*		450 k, EPIC	Novel diagnostic DNA methylation episignatures expand and refine the epigenetic landscapes of Mendelian disorders	[26]
	*DNMT3B*, *CDCA7*, *ZBTB24*, *HELLS*		450 k, EPIC	Functional correlation of genome-wide DNA methylation profiles in genetic neurodevelopmental disorders	[38]
Intellectual developmental disorder with seizures and language delay (IDDSELD)	*SETD1B*		450 k, EPIC	Novel diagnostic DNA methylation episignatures expand and refine the epigenetic landscapes of Mendelian disorders	[40]
	*SETD1B*		450 k, EPIC	Functional correlation of genome-wide DNA methylation profiles in genetic neurodevelopmental disorders	[38]
Intellectual developmental disorder, autosomal dominant 21	*CTCF*		EPIC	Identification of DNA methylation episignature for the intellectual developmental disorder, autosomal dominant 21 syndrome, caused by variants in the *CTCF* gene	[95]
Intellectual developmental disorder, X-linked, syndromic, Armfield type (MRXSA)	*FAM50A*		EPIC	Detection of a DNA methylation signature for the intellectual developmental disorder, X-linked, syndromic, Armfield type	[29]
	*FAM50A*		450 k, EPIC	Novel diagnostic DNA methylation episignatures expand and refine the epigenetic landscapes of Mendelian disorders	[40]
	*FAM50A*		450 k, EPIC	Functional correlation of genome-wide DNA methylation profiles in genetic neurodevelopmental disorders	[38]
*JARID2* neurodevelopmental syndrome	*JARID2*		EPIC	DNA methylation signature for *JARID2*-neurodevelopmental syndrome	[96]
	*JARID2*		EPIC	Functional Insight into and Refinement of the Genomic Boundaries of the *JARID2*-Neurodevelopmental Disorder Episignature	[97]
Kagami–Ogata syndrome (KOS14)	14q32.2	Imprinting	450 k	Genome-wide multilocus imprinting disturbance analysis in Temple syndrome and Kagami-Ogata syndrome	[98]
Kabuki syndrome	*KMT2D*		450 k	The defining DNA methylation signature of Kabuki syndrome enables functional assessment of genetic variants of unknown clinical significance	[99]
	*KMT2D*		450 k	CHARGE and Kabuki syndromes: gene-specific DNA methylation signatures identify epigenetic mechanisms linking these clinically overlapping conditions	[61]
	*KMT2D*		450 k	Interplay between histone and DNA methylation seen through comparative methylomes in rare Mendelian disorders	[94]
	*KMT2D*		EPIC	Episignature analysis of moderate effects and mosaics	[75]
	*KMT2D*		NGS	Comparison of methylation episignatures in *KMT2B*- and *KMT2D*-related human disorders	[76]
	*KMT2D*		450 k	Patients with a Kabuki syndrome phenotype demonstrate DNA methylation abnormalities	[100]
	*KMT2D*		450 k	Comprehensive evaluation of the implementation of episignatures for diagnosis of neurodevelopmental disorders (NDDs)	[101]
	*KMT2D*, *KDM6A*		450 k	Genomic DNA methylation signatures enable concurrent diagnosis and clinical genetic variant classification in neurodevelopmental syndromes	[45]
	*KMT2D*		450 k, EPIC	Diagnostic utility of genome-wide DNA methylation testing in genetically unsolved individuals with suspected hereditary conditions	[36]
	*KMT2D*, *KDM6A*		450 k, EPIC	Evaluation of DNA methylation episignatures for diagnosis and phenotype correlations in 42 Mendelian neurodevelopmental disorders	[26]
	*KMT2D*, *KDM6A*		450 k, EPIC	Novel diagnostic DNA methylation episignatures expand and refine the epigenetic landscapes of Mendelian disorders	[40]
	*KMT2D*, *KDM6A*		450 k, EPIC	Functional correlation of genome-wide DNA methylation profiles in genetic neurodevelopmental disorders	[38]
KBG syndrome	*ANKRD11*, *16p24.3*	Variants or deletions	EPIC	*ANKRD11* pathogenic variants and 16q24.3 microdeletions share an altered DNA methylation signature in patients with KBG syndrome	[102]
*KDM2B*-related neurodevelopmental disorder	*KDM2B*		EPIC	Delineation of a KDM2B-related neurodevelopmental disorder and its associated DNA methylation signature	[103]
	*KDM2B*		450 k, EPIC	Novel diagnostic DNA methylation episignatures expand and refine the epigenetic landscapes of Mendelian disorders	[40]
	*KDM2B*		450 k, EPIC	Functional correlation of genome-wide DNA methylation profiles in genetic neurodevelopmental disorders	[38]
Kleefstra syndrome	*EHMT1*, 9q34.3	Microdeletions	EPIC	*EHMT1* pathogenic variants and 9q34.3 microdeletions share altered DNA methylation patterns in patients with Kleefstra syndrome	[104]
	*EHMT1*		450 k, EPIC	Evaluation of DNA methylation episignatures for diagnosis and phenotype correlations in 42 Mendelian neurodevelopmental disorders	[26]
	*EHMT1*		450 k, EPIC	Novel diagnostic DNA methylation episignatures expand and refine the epigenetic landscapes of Mendelian disorders	[40]
	*EHMT1*		450 k, EPIC	Functional correlation of genome-wide DNA methylation profiles in genetic neurodevelopmental disorders	[38]
*KMT2C*-related syndrome	*KMT2C*		EPIC	Pathogenic variants in *KMT2C* result in a neurodevelopmental disorder distinct from Kleefstra and Kabuki syndromes	[105]
	*KMT2C*		450 k, EPIC	Evaluation of DNA methylation episignatures for diagnosis and phenotype correlations in 42 Mendelian neurodevelopmental disorders **(Not significant)**	[26]
	*KMT2C*		450 k, EPIC	Novel diagnostic DNA methylation episignatures expand and refine the epigenetic landscapes of Mendelian disorders	[40]
Koolen–De Vries syndrome	*KANSL1*, 17q21.31	Deletion	EPIC	A new blood DNA methylation signature for Koolen-de Vries syndrome: Classification of missense *KANSL1* variants and comparison to fibroblast cells	[106]
	*KANSL1*		450 k, EPIC	Evaluation of DNA methylation episignatures for diagnosis and phenotype correlations in 42 Mendelian neurodevelopmental disorders	[26]
	*KANSL*		450 k, EPIC	Novel diagnostic DNA methylation episignatures expand and refine the epigenetic landscapes of Mendelian disorders	[40]
	*KANSL1*		450 k, EPIC	Functional correlation of genome-wide DNA methylation profiles in genetic neurodevelopmental disorders	[38]
Leigh syndrome (LS)	*MORC2*		EPIC, EPICv2	Pleiotropic effects of *MORC2* derive from its epigenetic signature	[60]
Luscan–Lumish syndrome (LLS)	*SETD2*	1740 codon, Truncating	NGS	Epigenotype-genotype-phenotype correlations in *SETD1A* and *SETD2* chromatin disorders	[107]
	*SETD2*		EPIC	Interplay between histone and DNA methylation seen through comparative methylomes in rare Mendelian disorders	[94]
	*SETD2*		450 k, EPIC	Novel diagnostic DNA methylation episignatures expand and refine the epigenetic landscapes of Mendelian disorders	[40]
	*SETD2*		450 k, EPIC	Functional correlation of genome-wide DNA methylation profiles in genetic neurodevelopmental disorders	[38]
Lynch syndrome	*MLH1*	Epimutation	450 k	Primary constitutional *MLH1* epimutations: a focal epigenetic event	[108]
Menke–Hennekam syndrome	*CREBBP*, *EP300*	ZZ, TAZ2, ID4	EPIC	Menke-Hennekam syndrome delineation of domain-specific subtypes with distinct clinical and DNA methylation profiles	[109]
	*CREBBP*, *EP300*		450 k, EPIC	Novel diagnostic DNA methylation episignatures expand and refine the epigenetic landscapes of Mendelian disorders	[40]
	*CREBBP*, *EP300*	ID4	450 k, EPIC	Functional correlation of genome-wide DNA methylation profiles in genetic neurodevelopmental disorders	[38]
Mental retardation (autosomal dominant 23)	*SETD5*		450 k, EPIC	Novel diagnostic DNA methylation episignatures expand and refine the epigenetic landscapes of Mendelian disorders	[40]
	*SETD5*		450 k, EPIC	Functional correlation of genome-wide DNA methylation profiles in genetic neurodevelopmental disorders	[38]
Mental retardation (autosomal dominant 51)	*KMT5B*		450 k, EPIC	Evaluation of DNA methylation episignatures for diagnosis and phenotype correlations in 42 Mendelian neurodevelopmental disorders	[26]
	*KMT5B*		450 k, EPIC	Novel diagnostic DNA methylation episignatures expand and refine the epigenetic landscapes of Mendelian disorders	[40]
	*KMT5B*		450 k, EPIC	Functional correlation of genome-wide DNA methylation profiles in genetic neurodevelopmental disorders	[38]
Mental retardation (X-linked)	*BRWD3*, *ZNF711*, *UBE2A*, *SMS*, *PHF8/*/*		450 k, EPIC	Evaluation of DNA methylation episignatures for diagnosis and phenotype correlations in 42 Mendelian neurodevelopmental disorders	[26]
	*BRWD3*, *ZNF711*, *UBE2A*, *SMS*, *PHF8/*/*		450 k, EPIC	Novel diagnostic DNA methylation episignatures expand and refine the epigenetic landscapes of Mendelian disorders	[40]
	*BRWD3*, *ZNF711*, *UBE2A*, *SMS*		450 k, EPIC	Functional correlation of genome-wide DNA methylation profiles in genetic neurodevelopmental disorders	[38]
Mitochondrial disease (MD)	*MORC2*		EPIC, EPICv2	Pleiotropic effects of *MORC2* derive from its epigenetic signature	[60]
Mowat–Wilson	*ZEB2*		EPIC	Identification of the DNA methylation signature of Mowat-Wilson syndrome	[110]
*MSL2*-related NDDs	*MSL2*		EPIC	*MSL2* variants lead to a neurodevelopmental syndrome with lack of coordination, epilepsy, specific dysmorphisms, and a distinct episignature	[111]
Myopathy, lactic acidosis and sideroblastic anemia 2 (MLASA2)	*YARS2*		450 k, EPIC	Novel diagnostic DNA methylation episignatures expand and refine the epigenetic landscapes of Mendelian disorders	[40]
	*YARS2*		450 k, EPIC	Functional correlation of genome-wide DNA methylation profiles in genetic neurodevelopmental disorders	[38]
Neurodevelopmental disorder with or without autism or seizures (NEDAUS)	*CUL3*		EPIC	*CUL3*-related neurodevelopmental disorder: Clinical phenotype of 20 new individuals and identification of a potential phenotype-associated episignature	[112]
Neurodevelopmental disorder with coarse facies and mild distal skeletal abnormalities (NEDCFSA)	*KDM6B*		450 k, EPIC	Evaluation of DNA methylation episignatures for diagnosis and phenotype correlations in 42 Mendelian neurodevelopmental disorders **(Not significant)**	[26]
Nicolaides–Baraitser syndrome	*SMARCA2*		EPIC	New insights into DNA methylation signatures: *SMARCA2* variants in Nicolaides-Baraitser syndrome	[113]
	*SMARCA2*		450 k, EPIC	BAFopathies’ DNA methylation epi-signatures demonstrate diagnostic utility and functional continuum of Coffin–Siris and Nicolaides–Baraitser syndromes	[65]
	*SMARCA2*		450 k, EPIC	Diagnostic utility of genome-wide DNA methylation testing in genetically unsolved individuals with suspected hereditary conditions	[36]
	*SMARCA2*		450 k, EPIC	Evaluation of DNA methylation episignatures for diagnosis and phenotype correlations in 42 Mendelian neurodevelopmental disorders	[26]
	*SMARCA2*		450 k, EPIC	Novel diagnostic DNA methylation episignatures expand and refine the epigenetic landscapes of Mendelian disorders	[40]
	*SMARCA2*		450 k, EPIC	Functional correlation of genome-wide DNA methylation profiles in genetic neurodevelopmental disorders	[38]
Phelan–McDermid syndrome	*SHANK3*, 22q13.3	Large deletions	EPIC	DNA methylation epi-signature is associated with two molecularly and phenotypically distinct clinical subtypes of Phelan-McDermid syndrome	[114]
	22q13.3	Deletions	450 k, EPIC	Novel diagnostic DNA methylation episignatures expand and refine the epigenetic landscapes of Mendelian disorders	[40]
	22q13.3	Deletions	450 k, EPIC	Functional correlation of genome-wide DNA methylation profiles in genetic neurodevelopmental disorders	[38]
Pitt–Hopkins syndrome	*TCF4*		EPIC	DNA methylation episignature and comparative epigenomic profiling for Pitt-Hopkins syndrome caused by *TCF4* variants	[115]
Prader–Willi syndrome	*15q11–q13*	Hypermethylated *SNURF*, UPD (15) maternal, epimutation, deletion	450 k	Genome-wide methylation analysis in Silver-Russell syndrome, Temple syndrome, and Prader-Willi syndrome	[116]
Progressive supranuclear palsy (PSP—tauopathies)	17q21.31		450 k	An epigenetic signature in peripheral blood associated with the haplotype on 17q21.31, a risk factor for neurodegenerative tauopathy	[87]
PTBP1-associated syndrome	*PTBP1*		EPIC	*PTBP1* variants displaying altered nucleocytoplasmic distribution are responsible for a neurodevelopmental disorder with skeletal dysplasia	[117]
*PURA*-related neurodevelopmental disorder	*PURA*, 5q31.23	Loss of function, deletion	EPIC	Genome-wide epigenetic signatures facilitated the variant classification of the *PURA* gene and uncovered the pathomechanism of *PURA*-related neurodevelopmental disorders	[118]
Rahman syndrome	*HIST1H1E*		EPIC	Frameshift mutations at the C-terminus of *HIST1H1E* result in a specific DNA hypomethylation signature	[119]
	*HIST1H1E*		450 k, EPIC	Evaluation of DNA methylation episignatures for diagnosis and phenotype correlations in 42 Mendelian neurodevelopmental disorders	[26]
	*HIST1H1E*		450 k, EPIC	Novel diagnostic DNA methylation episignatures expand and refine the epigenetic landscapes of Mendelian disorders	[40]
	*HIST1H1E*		450 k, EPIC	Functional correlation of genome-wide DNA methylation profiles in genetic neurodevelopmental disorders	[38]
Recurrent constellations of embryonic malformations (RCEM)			EPIC	Identification of a DNA methylation episignature for recurrent constellations of embryonic malformations	[120]
Renpenning syndrome	*PQBP1*		EPIC	Identification of a DNA methylation signature for Renpenning syndrome (RENS1), a spliceopathy	[121]
	*PQBP1*		450 k, EPIC	Novel diagnostic DNA methylation episignatures expand and refine the epigenetic landscapes of Mendelian disorders	[40]
	*PQBP1*		450 k, EPIC	Functional correlation of genome-wide DNA methylation profiles in genetic neurodevelopmental disorders	[38]
ReNU syndrome	*RNU4-2*		EPIC, EPICv2	Dominant variants in major spliceosome U4 and U5 small nuclear RNA genes cause neurodevelopmental disorders through splicing disruption	[122]
	*RNU4-2*		EPICv2	Characterization of snRNA-related neurodevelopmental disorders through the Spanish Undiagnosed Rare Disease Programs	[123]
Rett syndrome	*MECP2*		450 k	Genomic DNA methylation signatures enable concurrent diagnosis and clinical genetic variant classification in neurodevelopmental syndromes **(Not significant)**	[45]
	*MECP2*		450 k, EPIC	Evaluation of DNA methylation episignatures for diagnosis and phenotype correlations in 42 Mendelian neurodevelopmental disorders **(Not significant)**	[26]
Rubinstein–Taybi syndrome	*CREBBP*, *EP300*		450 k, EPIC	Evaluation of DNA methylation episignatures for diagnosis and phenotype correlations in 42 Mendelian neurodevelopmental disorders	[26]
	*CREBBP*, *EP300*		450 k, EPIC	Novel diagnostic DNA methylation episignatures expand and refine the epigenetic landscapes of Mendelian disorders	[40]
	*CREBBP*, *EP300*		450 k, EPIC	Functional correlation of genome-wide DNA methylation profiles in genetic neurodevelopmental disorders	[38]
Saethre–Chotzen syndrome	*TWIST*		450 k	Genomic DNA methylation signatures enable concurrent diagnosis and clinical genetic variant classification in neurodevelopmental syndromes **(Not significant)**	[45]
Say–Barber–Biesecker–Young–Simpson syndrome (SBBYSS, Ohdo syndrome)	*KAT6B*		450 k	Genomic DNA methylation signatures enable concurrent diagnosis and clinical genetic variant classification in neurodevelopmental syndromes	[45]
	*KAT6B*		450 k, EPIC	Evaluation of DNA methylation episignatures for diagnosis and phenotype correlations in 42 Mendelian neurodevelopmental disorders	[26]
	*KAT6B*		450 k, EPIC	Novel diagnostic DNA methylation episignatures expand and refine the epigenetic landscapes of Mendelian disorders	[40]
	*KAT6B*		450 k, EPIC	Functional correlation of genome-wide DNA methylation profiles in genetic neurodevelopmental disorders	[38]
Schizophrenia			450 k	Screening for rare epigenetic variations in autism and schizophrenia	[50]
Scrap-related syndrome (non-specified)	*SRCAP*	Proximal variants	EPIC	Truncating *SRCAP* variants outside the floating-harbor syndrome locus cause a distinct neurodevelopmental disorder with a specific DNA methylation signature	[85]
*SETD1B*-related syndrome	*SETD1B*, *KMT2B*	Deletion, SNVs	EPIC	A genome-wide DNA methylation signature for *SETD1B*-related syndrome	[124]
	*SETD1B*		450 k, EPIC	Evaluation of DNA methylation episignatures for diagnosis and phenotype correlations in 42 Mendelian neurodevelopmental disorders	[26]
	*SETD1B*		450 k, EPIC	Novel diagnostic DNA methylation episignatures expand and refine the epigenetic landscapes of Mendelian disorders	[40]
Sifrim–Hitz–Weiss syndrome (SIHIWES)	*CHD4*	Nonsense, ATPase domain, PHD domain	EPICv2	Discovery of a DNA methylation profile in individuals with Sifrim-Hitz-Weiss syndrome	[125]
Silver–Russell syndrome (SRS)	*H19/IGF2*	Maternal UPD7 and ICR1	450 k	Genome-wide methylation analysis in Silver-Russell syndrome patients	[32]
	*H19/IGF2*	Loss of methylation	450 k	Genome-wide methylation analysis in Silver-Russell syndrome, Temple syndrome, and Prader-Willi syndrome	[116]
Smith–Magenis syndrome	*RAI1*		450 k, EPIC	Evaluation of DNA methylation episignatures for diagnosis and phenotype correlations in 42 Mendelian neurodevelopmental disorders **(Not significant)**	[26]
Sotos syndrome	*NSD1*		450 k	*NSD1* mutations generate a genome-wide DNA methylation signature	[126]
	*NSD1*		450 k	Comprehensive evaluation of the implementation of episignatures for diagnosis of neurodevelopmental disorders (NDDs)	[101]
	*NSD1*		450 k	Interplay between histone and DNA methylation seen through comparative methylomes in rare Mendelian disorders	[94]
	*NSD1*		450 k	Genomic DNA methylation signatures enable concurrent diagnosis and clinical genetic variant classification in neurodevelopmental syndromes	[45]
	*NSD1*		450 k, EPIC	Diagnostic utility of genome-wide DNA methylation testing in genetically unsolved individuals with suspected hereditary conditions	[36]
	*NSD1*		450 k, EPIC	Evaluation of DNA methylation episignatures for diagnosis and phenotype correlations in 42 Mendelian neurodevelopmental disorders	[26]
	*NSD1*		450 k, EPIC	Novel diagnostic DNA methylation episignatures expand and refine the epigenetic landscapes of Mendelian disorders	[40]
	*NSD1*		450 k, EPIC	Functional correlation of genome-wide DNA methylation profiles in genetic neurodevelopmental disorders	[38]
Spinal muscular atrophy (SMA)	*MORC2*		EPIC, EPICv2	Pleiotropic effects of *MORC2* derive from its epigenetic signature	[60]
*SRSF1*-related syndrome	*SRSF1*		EPIC	*SRSF1* haploinsufficiency is responsible for a syndromic developmental disorder associated with intellectual disability	[127]
Tatton-Brown–Rahman syndrome	*DNMT3A*		EPIC	Growth disrupting mutations in epigenetic regulatory molecules are associated with abnormalities of epigenetic aging	[128]
	*DNMT3A*		EPIC	Interplay between histone and DNA methylation seen through comparative methylomes in rare Mendelian disorders	[94]
	*DNMT3A*		450 k, EPIC	Evaluation of DNA methylation episignatures for diagnosis and phenotype correlations in 42 Mendelian neurodevelopmental disorders	[26]
	*DNMT3A*		450 k, EPIC	Novel diagnostic DNA methylation episignatures expand and refine the epigenetic landscapes of Mendelian disorders	[40]
	*DNMT3A*		450 k, EPIC	Functional correlation of genome-wide DNA methylation profiles in genetic neurodevelopmental disorders	[38]
Temple syndrome	14q32.2	Imprinting	450 k	Genome-wide multilocus imprinting disturbance analysis in Temple syndrome and Kagami-Ogata syndrome	[98]
	14q32.2	UPD14 maternal Epimutation	450 k	Genome-wide methylation analysis in Silver-Russell syndrome, Temple syndrome, and Prader-Willi syndrome	[116]
Unidentified cases (mixed congenital diseases)			450 k	Identification of rare de novo epigenetic variations in congenital disorders	[129]
Weaver syndrome	*EZH2*		450 k	DNA methylation signature for *EZH2* functionally classifies sequence variants in three PRC2 complex genes	[130]
	*EZH2*		450 k	Genomic DNA methylation signatures enable concurrent diagnosis and clinical genetic variant classification in neurodevelopmental syndromes (Not significant)	[45]
	*EZH2*		450 k, EPIC	Novel diagnostic DNA methylation episignatures expand and refine the epigenetic landscapes of Mendelian disorders	[40]
	*EZH2*		450 k, EPIC	Functional correlation of genome-wide DNA methylation profiles in genetic neurodevelopmental disorders	[38]
Werner syndrome	*WRN*		EPIC	Genome-wide DNA methylation analysis in blood cells from patients with Werner syndrome	[131]
	*WRN*, *LMNA*, *POLD1*		EPIC	Epigenetic signatures of Werner syndrome occur early in life and are distinct from normal epigenetic aging processes	[132]
Wiedemann–Steiner syndrome	*KMT2A*		EPIC	Clinical utility of a unique genome-wide DNA methylation signature for *KMT2A*-related syndrome	[133]
	*KMT2A*		450 k, EPIC	Evaluation of DNA methylation episignatures for diagnosis and phenotype correlations in 42 Mendelian neurodevelopmental disorders	[26]
	*KMT2A*		450 k, EPIC	Novel diagnostic DNA methylation episignatures expand and refine the epigenetic landscapes of Mendelian disorders	[40]
	*KMT2A*		450 k, EPIC	Functional correlation of genome-wide DNA methylation profiles in genetic neurodevelopmental disorders	[38]
Williams syndrome	7q11.23	Deletion	450 k	Symmetrical dose-dependent DNA-methylation profiles in children with deletion or duplication of 7q11.23	[37]
	7q11.23	Deletion	450 k	Integrated DNA methylation analysis reveals a potential role for *ANKRD30B* in Williams syndrome	[134]
	7q11.23	Deletion	450 k	Diagnostic utility of genome-wide DNA methylation testing in genetically unsolved individuals with suspected hereditary conditions	[36]
	7q11.23	Deletion	450 k, EPIC	Evaluation of DNA methylation episignatures for diagnosis and phenotype correlations in 42 Mendelian neurodevelopmental disorders	[26]
	7q11.23	Deletion	450 k, EPIC	Novel diagnostic DNA methylation episignatures expand and refine the epigenetic landscapes of Mendelian disorders	[40]
	7q11.23	Deletion	450 k, EPIC	Functional correlation of genome-wide DNA methylation profiles in genetic neurodevelopmental disorders	[38]
Witteveen–Kolk syndrome	*SIN3A*		EPIC	DNA methylation episignature for Witteveen-Kolk syndrome due to *SIN3A* haploinsufficiency	[135]
Wolf–Hirschhorn syndrome	4q16.13, *NSD2*	Deletion	EPIC	Loss of function in *NSD2* causes DNA methylation signature similar to that in Wolf-Hirschhorn syndrome	[136]
	4q16.13	Deletion	450 k, EPIC	Novel diagnostic DNA methylation episignatures expand and refine the epigenetic landscapes of Mendelian disorders	[40]
	4q16.13	Deletion	450 k, EPIC	Functional correlation of genome-wide DNA methylation profiles in genetic neurodevelopmental disorders	[38]

**Table 2 biomedicines-13-03043-t002:** Computational resources for episignature development used in any step of the pipeline, extracted from the literature.

Step	Package	Source	Usage	Ref.
Array processing	minfi	R Bioconductor	.idat reading and processing.	[146]
	ChAMP	R Bioconductor	.idat reading and processing.	[147,148]
	lumi	R Bioconductor	.idat reading and processing.	[149]
	SeSAMe	R Bioconductor	.idat reading and processing.	[150]
	meffil	R Bioconductor	.idat reading and processing.	[151]
	RnBeads	R Bioconductor	.idat reading and processing.	[152,153]
	Genome Studio	Illumina	.idat reading and processing (official Illumina tool).	
	IMA	R	.idat reading.	[154]
	MethylCallR	R Bioconductor	.idat reading and processing.	[155]
	MethAid	R Bioconductor	.idat reading and processing.	[156]
	wateRmelon	R Bioconductor	.idat reading and processing.	[157]
	maxProbes	R	CpG QC filters.	
	sva	R	Surrogate variable identification and batch correction.	[158]
NGS processing	Trim Galore!	Babraham Bioinformatics (Bash)	FASTQ quality control.	
	Bismark	Babraham Bioinformatics (Bash)	Bisulfite sequencing aligner and methylation caller.	
Epigenetic analysis and episignature development	RnBeads	R Bioconductor	Methylation processing data. Allow .bismarkCov files.	[152,153]
	limma	R Bioconductor	Array data analysis.	[159,160]
	ChAMP	R Bioconductor	Array data analysis.	[147,148]
	meffil	R Bioconductor	Array data analysis.	[151]
	Qlucore	Web-based	Methylation analysis.	
	minfi	R Bioconductor	DMP analysis. Blood cells proportion estimation (Houseman). DMR analysis (bumphunter).	[146]
	FlowSorted. Blood.EPIC	R Bioconductor	Blood cells proportion estimation.	[161]
	EpiDISH	R Bioconductor	Blood cells proportion estimation.	[162]
	DMRcate	R Bioconductor	DMR analysis.	[163]
	Golden Helix—SNP and variation	Independent app.	DMP analysis.	
	MedCalc	Independent app	DMP analysis.	
	WGCNA	R	Weighted correlation network analysis. Identification of co-methylation system networks.	[164]
	bumphunter	R Bioconductor	DMR analysis.	[165]
	comp-p	R	DMR analysis.	[166]
	mRMRe	R	Feature selection through minimum-redundancy-maximum-relevance ensemble algorithm.	[167]
	DNAmAge	Web-based	Epigenetic age estimation (Horvath clock).	
	methylclock	R Bioconductor	Epigenetic age estimation.	[168]
Annotation and enrichment	Illumina annotation packages	R Bioconductor	CpG annotation for all Illumina arrays.	
	annotatr	R Bioconductor	Region annotation.	[169]
	LOLA	R Bioconductor	Region annotation.	[170]
	missMethyl	R Bioconductor	CpG enrichment analysis.	[171,172]
	methylGSA	R Bioconductor	CpG enrichment analysis.	[173]
	GSEA	R	Gene enrichment analysis.	[174]
ML model training	Caret	R	Train machine learning models. Offers wide range of customization, balancing, and CV techniques.	
	Kernlab	R	Train support vector machines.	[175]
	e1071	R	Train support vector machines	
	CALF	R	Train coarse approximation linear function.	
	glmnet	R	Train ridge, lasso and elastic net regression.	
	scikit-learn	Python	Train machine learning models. Offers wide range of customization, balancing and CV techniques.	
	GenPipes	Bash	NGS workflow management.	[176]

## Data Availability

No new data were created or analyzed in this study. Data sharing is not applicable to this article.

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
