# Peer review of "Analysis Methods for Diagnosing Rare Neurodevelopmental Diseases with Episignatures: A Systematic Review of the Literature"

_biomedicines, 2025, doi:10.3390/biomedicines13123043_

Round 1

Reviewer 1 Report

Comments and Suggestions for Authors

This article by Albert Alegret-García and co-authors provides an extensive and thorough review of the literature on episignatures and epimutations, corresponding platforms, tools, methods to both discover new episignatures and epimutations as well as to interrogate existing models. The paper is very well written overall and can interest a vast spectrum of readers. The figures and introduction will be accessible and very helpful to anyone wishing to join the field. Experts will rejoice in all the details. The literature on the subject is so vast and effervescent that the tables, supplementary tables and comparison of methodologies will be immensely useful to anyone in the field wishing to remain fully up to date and discover new ways to improve their data processes.

The only limitation I see to the manuscript is the “results” section. Compared to the rest of the manuscript, this section is particularly hard to read. I understand that the exercise of summarizing an extensive list of peculiarities, of varying methods and tools at each different step is a really difficult task. However, it would help the reader to identify some important messages to structure the section.

I have some minor remarks to moderate or clarify some of specific points, if the authors agree.

1- Figure 1: there is a typo, “casual” instead of “causal”

Besides this typo, I find the “no causal variant identified” category vague. If the review is in part intended for beginners in the field, then it would help to stress out that the “no causal variant identified” can either be VUS testing (variant identified, but causality into question) or clinical hypothesis confirmation (clinical points towards a given syndrome, but as the authors said syndromes share overlapping traits, so there is a possibility of mistake). These are very different objectives with different chances of success regarding episignatures. Sometimes there is even the possibility of running “blind” interrogation of the whole set of signatures.
Could you clarify this diversity of cases in the figure (if you can do it while keeping it clear)?

2- Figure 4: The advent of cross-validation or external validation, which are too often lacking in episignatures, despite the strong risk of overfitting, is not mentioned in this figure. I think it is important to remind readers of the risk and the importance of validation in this figure, to present it as an important step. If it is missing from this figure, then it could seem optional to the reader.

Could you add this?

3- l.243 “due to restrictions in genetic data sharing”. This is the common reason given by EpiSign but this cannot be the real reason why the data are not shared. Many 450K and Epic datasets have been shared by other groups. They would not if there had been ethical reasons against it. It is not the place of this review to copy-paste this false reason (hiding purely commercial motives).

Could you moderate/correct this assertion?

4- l.422-429 I understand it is hard to talk in advance of validation while at pre-processing steps. The authors do mention that “In the end, the researcher must verify whether batch correction eliminated the batch and preserved the expected biological variation.” They also encourage readers to look for inflated differences. All this is very important (As the authors said l.404 “batch correction is critical“). Indeed, batch effects do not only mean loss of power, it also implies type 1 error rate inflation and risk of publishing an episignature that will not reproduce on other data. I think it is important to be as practical on this aspect as you managed to be everywhere else in the manuscript.

Could you shortly remind non expert statisticians and beginners in the field of several practical tools and strategies available to check the validity of batch correction? For instance, define how to check for 1st error rate inflation? Stress the importance to resort to cross-/internal/external validation at the end to validate the absence of batch effects resulting in overfitting, etc. (possibly by linking the final paragraphs on validation)

5. l.489 and following. Beware of the dates of publication. You are here discussing old articles (in the timeline of episignature literature), with old methodology. I am not sure the same authors would use the same strategy if they had the opportunity to have a second chance at it.

Could you moderate this part? Or adopt a chronological view on the evolution of methodologies?

6. l.506 This particular paragraph is very hard to follow, to see where we are going. The main message here appears to be “everyone does what they want and no article is the same”. Is it really where you want to go? Some methods are different in appearance or name but are to be equivalent. The real difficulty is that perhaps teams resort to different methodologies because to come up with some significant signals they have to test and try several strategies and finally publish the winner? This strategy is highly risky, because failure to admit multiple testing and prior analysis schemes might/will lead to overfitting and lack of reproducibility. This means that for most signatures we do not know whether the signature is robust (to dataset and methods) or it could only be seen on this particular dataset with this methodology.

Could you find an angle to structure this paragraph (in relation to the whole result section)?

l.631-642 “likely reflecting disease-specific epigenetic alterations that require tailored approaches”. I really doubt it. Admittedly, the threshold is higher for Sotos, because half the genome would light up on differential analysis. The effect is so strong that a whole genome PCA separates Sotos patients from controls without requiring the prior-identification of a particular subset of DMPs. The threshold is augmented to reduce the number of positions to feed into the SVM. For syndromes with weaker “disease-specific epigenetic alterations”, the thresholds are just traditionally reduced until a reasonable enough number of positions get significant depending on the strength of methylation differences but also and foremost on sample size. So, it is in part linked to the syndrome-specific strength of methylation alterations but also strongly linked to the statistical power and sample size that authors managed to gather and their subjective idea of a “credible signature size”. I fear the reality is less scientific and biologically meaningful that the authors let it be.

Could you rephrase/moderate this sentence?

l.778-779 “By and large, we observed classification models of rare diseases based on episignatures reported very good accuracy, regardless of the method used, the disease studied or the screening model.“

In truth, it is difficult to say when there has been no external validation for most of the 100 and so episignatures. As you emphasize, the methodology is often tuned to the dataset to reach sufficient enough predictive properties for publication. There must be some winner’s curse and over-estimation of the real predictive performances, at least for some episignatures. You cited Husson et al, but Methadory authors also pinpoint difficulties in replicating some of the episignatures. It seems to me that this sentence is overly optimistic. True for most, perhaps, but not for all. Not until proven so by careful external validation.

Could you rephrase/moderate this sentence?

l.800 The authors cite Datashield but also the European Genome-phenome Archive (EGA) also provides secure sharing and access of methylation data, possibly under the protection of DTAs.

Could you add the reference as well ?

Author Response

Please see the attachment. Manuscript changes commented in the .pdf version.

Reviewer 2 Report

Comments and Suggestions for Authors

This complex and long review provide a comprehensive framework for episignature-based diagnosis of NDDs based on recent references, to guide clinical practitioners and researchers in the field.
The goal of this text is important and timely. The authors however should improve their text by a multiple number of factors. Here the significant ones
- wording is too much technical for the reader of the Journal and the description of the many steps in the analytical pipeline are full of jargons not familial to geneticists
- the long list of disorders is of limited use in the text. I see more appropriate moving to appendix or supplemental material
- most figures are of limited quality for precise revision. Please provide better quality figures
- texts can often be replaces by schemes or figures to detail the steps in diagnosis. At time, this manuscript sounds like a technical manual not a research article. Please modify significantly the layout of the results section and the Discussion
- limitations in the review are important and should be included at the end of Discussion section
- in the Conclusion offer an insight on how useful this approach will be in the assessing diagnostic rates in NDDs and in other fields of neurogenetics

Author Response

(The authors gave the same response as above.)

Reviewer 3 Report

Comments and Suggestions for Authors

Remove the repeated sentence- line 75

This is a well-written review that offers valuable insights into the diagnosis of undiagnosed rare diseases

Are there any publications discussing undiagnosed rare diseases that do not involve genetic changes or CNVs but focus solely on methylation defects aside from BWS and PWS? If such cases exist, can these methylation defects be associated with disease? Could they be used to investigate genes with abnormal methylation patterns to diagnose a new rare disease? Additionally, how are the epimutations inherited?

How is counselling offered to the families? 

Is there a relationship between methylation, DNA damage, and micronuclei? If so, can this relationship be utilised to develop a cost-effective method for detecting methylation status?

Author Response

(The authors gave the same response as above.)

Round 2

Reviewer 2 Report

Comments and Suggestions for Authors

No further questions